# A survey on the attitudes of parents with young children on in-home monitoring technologies and study designs for infant research

Laurel A. Fish[1]*, Emily J. H. Jones[2]

1 Institute of Psychiatry, Psychology and Neuroscience, King's College London, London, United Kingdom,
2 Centre for Brain and Cognitive Development, Birkbeck, University of London, London, United Kingdom

☉ These authors contributed equally to this work.
* laurel.fish@kcl.ac.uk

## Abstract

Remote in-home infant monitoring technologies hold great promise for increasing the scalability and safety of infant research (including in regard to the current Covid-19 pandemic), but remain rarely employed. These technologies hold a number of fundamental challenges and ethical concerns that need addressing to aid the success of this fast-growing field. In particular, the responsible development of such technologies requires caregiver input. We conducted a survey of the opinions of 410 caregivers on the viability, privacy and data access of remote in-home monitoring technologies and study designs. Infant-friendly wearable devices (such as sensing body suits) were viewed favourably. Caregivers were marginally more likely to accept video and audio recording in the home if data was anonymised (through automated processing) at point of collection, particularly when observations were lengthy. Caregivers were more open to international data sharing for anonymous data. Caregivers were interested in viewing all types of data, but were particularly keen to access video and audio recordings for censoring purposes (i.e., to delete data segments). Taken together, our results indicate generally positive attitudes to remote in-home monitoring technologies and studies for infant research but highlight specific considerations such as safety, privacy and family practicalities (e.g. multiple caregivers, visitors and varying schedules) that must be taken into account when developing future studies.

## Introduction

Our burgeoning understanding of neurocognitive development has been made possible by the development of novel technologies [1–3]. Often requiring a team of trained specialists working in highly controlled testing environments [4,5], these measures are typically administered in university babylabs [4,6]. Although there are a number of merits to lab-based experimental designs, such as the control of testing environments and structured assessments to elicit behaviours of interest [7,8]; there are broadening debates over the ecological validity of the measures

**Data Availability Statement:** All data files are available from the OSF database (osf.io/fbhd8).

**Funding:** This collection of data has been supported by a grant from the European Community's Horizon 2020 Program (https://ec.

europa.eu/programmes/horizon2020/en) under grant agreement n° 642990 (EJ), and the Innovative Medicines Initiative Joint Undertaking (https://www.eu-aims.eu/imprint/) under grant agreement n° 115300 (EJ), resources of which are composed of financial contribution from the European Union's Seventh Framework Programme (FP7/2007 - 2013) and EFPIA companies' in kind contribution. The results leading to this project has been supported by AIMS2 (https://www.aims-2-trials.eu), which received funding from the Innovative Medicines Initiative 2 Joint Undertaking under grant agreement No 777394 (EJ). This Joint Undertaking receives support from the European Union's Horizon 2020 research. The funders had no role in study design, data collection and analysis, decision to publish, or preparation of the manuscript.

**Competing interests:** The authors have declared that no competing interests exist.

obtained. Babylabs are unnatural settings [6,7,9], which may distort the targeted natural behaviour under observation [10]. Additionally, babylabs can only facilitate relatively small windows of data collection, and in longitudinal designs this can result in data points scaled months apart. Researchers are unable to capture the range of environments infants experience during development, limiting the range of measurable behaviours [7,8]. Finally, testing in babylabs across the world has recently been suspended due to the Covid-19 pandemic; this disruption to data collection will be particularly damaging for longitudinal cohorts.

A more ecologically valid, representative and Covid-safe alternative is in-home monitoring of an infant's daily activity using remote technology [8]. In addition to allowing the measurement of an infant's behaviour in their natural environment, such approaches allow longer and more frequent epochs of recording. These provide the opportunity to capture a range of behaviours, including those that happen rarely [11] (e.g. a child's first steps) and/or infrequently [8] (e.g. tantrums). Such techniques can capture more accurately the temporal structure and variability of dynamic interacting behaviours and environments [8,12]. Similarly, longer measurement periods may also shed new light on behaviours that have traditionally been measured over relatively short epochs of data collection [12,13]. Additionally, as remote technologies can be implemented by caregivers, they also provide an opportunity for socially distanced data collection at a more scalable level and may facilitate the participation of families who would find it difficult to come to a lab-based study (e.g. those with disabilities, working caregivers or remotely located families).

Commercially, remote home-monitoring technology for infants is not a new concept, with an abundant and expanding range of devices targeted towards and readily accepted by caregivers [14]. User-friendly, commercial devices do not allow access to raw, high resolution data and are therefore poorly equipped for more complex research questions [14]. For research, customized devices are preferable, where a team of multidisciplinary experts manufacture measurement tools with research-grade functionality that are also usable and aesthetic [8]. Such devices include baby bodysuits woven with sensing threads (herein refered to a smart suits; [15–17]), sensing ankle/wrist bands [18,19], sensing sticker electrodes [20–22], home camera/audio devices [23,24] and smartphone apps [25,26]. Such remote devices are providing unique research potential for in-home infant monitoring. For example, using a specially designed infant baby-grow with a built in actigraph, GPS and microphone, researchers demonstrated, in the home, extreme low and high levels of ambient noise are associated with reduction in spontaneous movement in infants [27].

Despite the growing promise, the implementation of home-monitoring tools in developmental research is still limited to a small number of studies with relatively small populations [28]. One critical area that has received limited attention is the views of primary caregivers on these tools. A user-centered model is essential for the development of first-rate technology which elicits minimum caregiver burdens and sacrifice [29,30]. Equally, determining caregiver opinions on this technology alongside the context of possible assessment designs is important. This knowledge is critical to optimise the successful development and broader deployment of remote monitoring technologies as well as shaping the possible questions and designs that researchers are able implement [28]. To our knowledge there have been few systematic attempts to canvass caregiver opinions on remote infant monitoring technology in the context of potential study designs on a UK cohort. In our view, to gauge acceptance of UK caregivers feedback is most critical in three areas; tool viability, privacy and data access.

## Viability

First, optimising the *viability* of tools and assessment designs so they are deemed acceptable, practical and minimally disruptive by caregivers is necessary for ensuring engagement and

uptake in future investigations. In terms of viability, remote tools and assessments can be categorised by the required amount of active care [8]. For example, when measuring heart rate, having to carefully apply and remove adhesive sticker sensing electrodes requires a lot more active care relative to using sensing smart suit or bands. These can be easily and quickly put on/taken off the infant and are not too far removed from daily dressing routines, thus are low active care devices. However, currently available sensing suits and bands do not provide sufficient data quality, particularly for physiological data where sensors often require good contact with skin and are prone to movement artefacts [31,32]. Similarly, video and audio devices can be bracketed into levels of active care. Static devices (e.g. those attached to cot or ceiling) require a lot less active care than on-body devices, which require charging or general maintenance (e.g. placing on infant and positioning). Although merited, static devices can often restrict research designs, with the need for careful consideration as to the spatial location of the behaviour of interest [8]. LENA [33], a lightweight wearable audio device, is already showing high feasibility and acceptance having been deployed in a large number of studies in the field of infant development [1]. Despite this, caregivers' opinions on static low active care versus on-body high active care video devices is unclear. Knowledge of parent's opinions of different remote devices with differing levels of required active care will equip researchers to consider the trade-off between quality/richness and viability–an important consideration for more lengthy or longitudinal studies where attrition is an inherent concern.

The contextualisation of data is another key requirement of remote designs (e.g. physiological measures that do not provide identifiable data). In a lab setting, environments are controlled, but in a home setting the context in which the data is being collected can vary dramatically both between infants (e.g. single child, multi-generational household, etc) and within an infant across a day (e.g. dinner time, playtime, bedtime etc). Recording this contextual information is sometimes required in order to understand the data (e.g. did an infant's heartrate increase because they were crying or laughing?). Contextualising information could be collected using smartphones via self-reporting [34]. For example, a caregiver could input their infant's current activity in response to a prompt. The increasing societal ubiquity of smartphones [35] make them an ideal tool for the collection of both contextualisation data and primary data [35,36], where timely notifications for question responses may reduce biasing retrospective reporting. Furthermore, many infant sensing devices in the commercial market already utilise smartphone apps for end-user device interaction-platform [37]. Utilising smartphones alongside wearable sensing technologies hold potential to enrich remote monitoring datasets, making them more informative. Understanding caregivers' opinions on the use of smartphones for interacting with remote infant monitoring studies will gauge acceptability for future investigations.

## Privacy

Using remote technologies for infant monitoring protocols in the home raises critical questions concerning privacy, particularly regarding transmission, storage and analysis of data. Such concerns of privacy are in part covered by legal frameworks such as the General Data Protection Regulation (GDPR) in the EU (2016/679; [38]), but the additional oversight of local and national ethics boards should be informed by the opinions of caregivers.

These privacy concerns are particularly relevant for non-anonymous, highly identifiable data (e.g. video and audio). A recent report on a USA-based sample of parents indicated privacy-preserving techniques minorly improved willingness to participate in the collection of identifiable data on their infant [28]. Such privacy-preserving techniques included the implementation of computer algorithms to automatically extract measures of behaviour and remove

identifiable information [28]. However, this research did not investigate the impact of privacy-preserving techniques on willingness to participate across different assessment structures (e.g. across differing periods/times of day) [24]. Considering how participants view the efficacy of privacy-preserving measures in the context of different research designs will enable researchers to make informed decisions on the trade-off between data quality and participant uptake/attrition.

Data sharing is another issue bracketed within privacy. Data sharing is encouraged by the Open Data Movement [39], a campaign to render datasets accessible and reusable. This is useful for improving the transparency and reproducibility of original data, and can facilitate inter-disciplinary and new research [40]. To adhere with American Psychological Association research standards [41] and General Data Protection Regulations [38] consent for data sharing must be taken explicitly. However, it is critical to gather caregiver views on their degree of comfort with different types of data sharing, particularly across data with differing levels of anonymity (e.g. raw data or those that have been anonymised through aforementioned privacy-preserving techniques). A previous survey on caregivers opinions established that anonymisation of data produced a weak non-significant increase in willingness to share [28]. However, this study was conducted in the US and did not gauge the willingness to share data across different geographic areas, which will ultimately determine the degree to which data is "open" within the research community.

## Data access

One potential application of intense in-home data collection is to identify patterns of development. This raises the question of *data access*–to what extent do caregivers want to access the data collected on their child, and in what format can and should that be provided? In lab-based studies it is common for parents to have no access to collected data unless official requests are made (e.g. through the Data Protection Act in the UK). However, this may not be desirable when considering more intense multi-device remote home-based approaches. This is particularly relevant considering previous investigations have established a 66–67% increase in willingness to share sensitive data when provided the opportunity to receive access to summary data [28]. However, it remains unclear as to whether caregivers value data access because of curiosity about their infant's development, or whether they want to be able to check over data streams from more intrusive measures (e.g. video data). The latter is particularly relevant when considering both ethical rights for participant withdrawal/retrospective data redaction as well as EU GDPR's right to erasure of identifiable data [38]. Gauging reasons for desiring data access (e.g. for viewing or censoring) across data with differing levels of anonymity will enable researchers to consider the pro/cons as well as the format of data access they should provide in their studies.

## Study goals

The goal of our study was to canvass the opinions of UK caregivers on the use of remote monitoring technologies in order to provide guidance for future research studies employing these approaches in similar UK based labs/research groups. Specifically, we developed an online survey to gauge the attitudes of caregivers of young children/infants towards a range of home-based technologies and assessment designs for research into infant psychophysiology/motor and behavioural development. Our questions were delivered in the context of real-world examples of such technologies and assessments to provide a concrete framing in which to ascertain realistic opinions from responders. We addressed our core themes, operationalising

preferences through the specific metrics of likelihood, practicality and duration/time of participation.

## Viability

- **Sticker sensing electrodes versus smart suits:** For the collection of infant psychophysiological/motor data, are lower active care infant wearable technologies (e.g. smart suits) preferable to more traditional higher active care technologies (e.g. sensing electrode stickers)?

- **Static versus body image-only video recording devices:** Are static low active care video recording devices preferable to all-encompassing high active care on-body video recording devices?

- **Smartphones:** To what degree would caregivers wish to interact with researchers via smartphone technology?

## Privacy

- **Video and audio recording with/without privacy preservation:** For the collection of more highly identifiable data, do caregivers prefer image-only video or audio recording of their infants, and how do caregivers view privacy-preserving automatic pre-processing versus manual non-anonymising pre-processing of video/audio data?

- **Data sharing:** Does type of technology alter who caregivers are willing to share their data with?

## Data access

- What types of data access do caregivers want?

## Future participation

- What extent are caregivers interested in the participation of remote in-home infant monitoring studies?

## Methods

### Recruitment procedure

We distributed the survey (See S1 File) to participants via two main streams over 6 months. First, an invitation containing a weblink was emailed via the SurveyMonkey Platform (surveymonkey.com), a public online platform for survey distribution and response collection. We emailed 1061 Birkbeck University of London's BabyLab UK-based database volunteers. Participants are recruited into this database via word of mouth, magazine adverts, social media and search engines. All members of the database had previously consented for such contact, making them a relevant sample as they would likely to participate in future studies incorporating the topics mentioned in the survey. Weblinks were also shared via social media (Twitter and Facebook) accounts belonging to Birkbeck University of London's BabyLab. In both recruitment methods, weblinks redirected responders to a webpage containing the survey debrief and instructions. Consent was indicated via tick box, with non-consenters redirected

to an "end of survey" webpage to close manually. Participants received no monetary compensation for their time.

## Participants

In total 513 people clicked on the weblink that directed them to the survey. Eight people declined consent and a further 95 people consented but did not answer any questions, therefore were not included in analysis. In total, we retained data from 410 individuals, with 333 individuals completing the entire survey (See S2 Table 1 and S2 Fig 1 for attrition rates in S2 File).

We developed the survey for the purpose of this study (See S1 File for the full non-copyright survey). The survey consisted of 61 questions (8 sections), which took an average of 12 minutes 18 seconds to complete. First, all responders provided basic information about their youngest child (age, gender and ethnicity) and family (parent education, household income, family health). Sections 2–7 asked about the responder's attitudes to the following technologies: smart suits, sensing electrode stickers, wrist/ankle bands, image-only video recording, audio recording, and smartphones (see S3 Table 1 for summary of technologies in S2 File). Section 8 asked attitudes to data access and sharing. Finally, we asked whether responders would be interested in future participation in a similar study.

The survey consisted of multiple-choice questions on the following topics: a) the likelihood of participation in studies with different designs; b) the practicality of technologies over different time scales (to assess overall practicality while accounting for the possible effect of length of study on practicality); c) the optimum length/time of participation; d) contact preference during the study (e.g. acceptable number of smartphone prompts). Likert style response options were offered to responders for questions corresponding to likelihood of participation (5 options: "Not at all", "Slightly", "Moderately", "Very", "Extremely") and practicality (4 option "Not at all", "Somewhat Impractical", "Somewhat Practical", "Practical"). Nominal multiple-choice options were offered for all other questions. To provide insight into unprecedented opinions, optional general comment boxes were provided in most sections for open-ended responses. Before being distributed, the survey was piloted on a small group of caregivers and adjusted accordingly. Once scripted, the survey was made available using SurveyMonkey.

## Ethical approval

Ethical approval was provided by Birkbeck University of London Research Ethics Committee (ID: 161776).

## Analysis

For each multiple-choice question, participants were given the option "prefer not to answer", which were considered as missing data in the analysis. For each comparative analysis, missing data were pairwise deleted. The Likert style scale responses were transformed to ordinal scales starting at 1 for the least favourable response and ending with 4 (practicality) or 5 (likelihood) for the most favourable. Statistical analysis was conducted using SPSS (IBM version 25.0.0.1; [42]) and R (version 3.5.1; [43]) using non-parametric tests due to the ordinal and categorical nature of the data. We used Leximancer Desktop 5.0 [44] to generate concepts and themes from open field answers for each general comment box. This analysis was not intended to be exhaustive, but to highlight themes beyond those considered when designing the survey. The Leximancer's concept map tool was utilised to display topographically the prevalence and co-occurrence of generated themes/concepts within the text (See S4 for more methodological

detail on Leximancer in S2 File). Analysis was divided into our three core themes (discussed above) assessed in the context of a set of technologies that, in our minds, are furthest advanced for remote infant monitoring research.

**Viability.**    One important factor influencing viability is the type of technological approach employed. We examined participant ratings to questions regarding acceptability and practicality of the remote collection of infant data using a) sticker sensing electrodes versus smart suits, b) static versus body image-only video recording devices, and c) smartphones.

*Sticker sensing electrodes versus smart suits*. We compared ratings given to sticker sensing electrodes and smart suits for questions regarding the following: a) likelihood of participating with each technology; b) practicality of using each technology for a "Short Period" or "Extended Period"; c) preferred length of participation for each technology. We used Wilcoxon Match-Pairs Signed-Ranks Test (WMPSR) to analyse the effect of technology on changes to likelihood rating. We employed a Friedman analysis of Variance (FANOVA) with pairwise Dunn-Bonferroni test to investigate differences in practicality responses to the four technology/length of study combinations (smart suits for "Short Period", smart suits for "Extended Period", sticker sensing electrodes for "Short Period", sticker sensing electrodes for "Extended Period". Using Chi-Square test with Bonferroni corrected binominal pairwise comparisons, we analysed whether, for smart suit and sticker electrodes invidually, responses to preferred length of participation were evenly distributed across time intervals ("Month", "Week", "Weekend", "Now and then", "Once", "Not at all"). Finally, to investigate whether preferred time interval for participation length differed between smart suits and sticker sensing electrodes, we used Bhapkar tests with 2*2 contingency Bonferroni corrected McNemar post-hoc test to compare selecting versus not selecting each time interval. Separate Leximancer concept/theme maps were generated to depict key themes and concepts noted within open-ended response boxes for each technology. Similar analysis comparing smart suits and sensing electrode stickers to wrist/ankle bands was also conducted (Methods and Results are reported in S5 in S2 File).

*Static versus body image-only video recording devices*. To examine acceptability of image-only video recording with differing levels of active care, we compared likelihood ratings given to static cot cameras with low active care or high active care body cameras. We conducted a Wilcoxon Match-Pairs Signed-Ranks Test (WMPSR) to determine the effect of video recording device on changes to likelihood rating.

*Smartphones*. To examine the acceptability of smartphones for the remote assessment of infants, we analysed responses to questions regarding: a) likelihood of using smartphones; b) practicality of using smart for a "Short Period" or "Extended Period"; c) preferred response time; c) preferred number of prompts. We employed One-Sample Wilcoxon Signed Rank Tests (O-SWSRT) to examine the degree to which the sample median of likelihood answers differed from the response median (2.5). We analysed how opinions on practicality changed for different durations of app interaction using WMPSR. We then analysed preferred response time ("Morning, "Afternoon", "Evening", "Anytime" or "Never") and prompt number ("Six", "Four", "Two", "One", "None" or "No Limit") using Chi-square with Bonferroni corrected binominal pairwise comparisons to determine if frequency of responders per response option significantly differed. A Leximancer concept/theme map was generated to depict open-ended response regarding smartphones.

**Privacy.**    In-home remote monitoring of infant participants raises privacy issues. We examine caregiver ratings to questions regarding a) video and audio recording with/without privacy preservation and b) data sharing.

*Video and audio recording with/without privacy preservation*. We compared ratings of video and audio recordings with and without privacy preserving pre-processing for questions

regarding a) likelihood of participation, b) preferred length of participation, and c) when they would be willing to be recorded. We also compared ratings on practicality of participating with video or audio recording for a "Short Period" or "Extended Period". We used Friedman Analysis of Variance (FANOVA) with pairwise Dunn-Bonferroni test to analyse whether: a) different recording device/processing option combinations affect participation likelihood ratings; and b) different recording device/recording duration combinations affect practicality ratings. Within each recording device/processing option combination, we conducted Chi-square with Bonferroni corrected binomial pairwise comparisons to determine whether participants' preferences were evenly distributed across participation length intervals ("Month", "Week", "Weekend", "Now and then", "Once", "Not at all") and recording time intervals ("Day and Night", "Day", "Night", "Specific time", "Never"). Using Bhapkar tests with Bonferroni corrected McNemar pairwise comparisons, we examined the effect of technology and processing method on a) preferred participation length and b) time of recording. Separate Leximancer concept/theme maps were generated to depict themes within free text comments for video and audio recording.

*Data sharing.* To determine responders' opinions on sharing data with research teams, we assessed responses to questions asking who responders would be willing to share their data with. As the survey question allowed multiple responses per individual, we considered the most international answer as the preferred data sharing option ("Across the world" and "All" were considered as the same response "International"). Chi-squared goodness-of-fit tests were employed to determine whether the frequency of data sharing preference was evenly distributed across different research teams (e.g. "International", "Europe", "UK", "Babylab" or "None") for a) anonymous data (e.g. heart rate) and b) non-anonymous data (e.g. video recording). Using Bhapkar tests with Bonferroni corrected McNemar pairwise comparisons, we also investigated whether participants responses to likelihood of sharing data with different research teams varied between the degree of identifiability of the data format.

**Data access.** Understanding whether participants may want to view their data and/or censor by accepting/deleting their recordings is crucial because it raises important questions over the appropriate types of data access researchers need to plan to provide. This could have significant resource and ethical implications in future investigations, as well as a potential impact on data redaction or participant withdrawal. To determine what types of data access caregivers wanted for each type of data, we examined responses to questions regarding accessing data for the purpose of a) viewing and b) censoring. We used Cochran's Q test with Dunn-Bonferroni pairwise comparisons to determine if the frequency of responders wanting to view or censor data significantly differed between technologies.

**Future participation.** Finally, we investigated interest and eligibility to participate by conducting Chi-Square Goodness-of-Fit analysis with Binomial pairwise comparisons. We also asked whether caregivers would be happy to use clean suits other families have used (S12 Note A in S2 File). Separate Leximancer concept/theme maps were generated for open-ended response. These results will provide an overall estimate of the likely scalability of such research approaches.

## Results

### Sample demographics

Throughout the survey, we asked responders to consider their youngest child (*Mean age* = 29.90 months, *SD* = 27.66 months) when answering questions. Of the 410 responders, 393 were the child's mother, 16 the father and one the grandparent. Sample demographics are presented in Table 1. See S6 for additional demographics on parent health and primary caregivers in S2 File.

**Table 1. Sample demographics.**

|  | % |  |
|---|---|---|
| **Child's Gender** |  |  |
| Female | 51.20 |  |
| Male | 48.00 |  |
| Non-binary | 0.20 |  |
| **Child's Ethnicity** |  |  |
| White | 80.24 |  |
| Black, Asian or Minority Ethnic Group | 19.02 |  |
| **Household Annual Income** |  |  |
| < £20,000 | 3.80 |  |
| £20,000 - £29,000 | 8.59 |  |
| £30,000 - £39,000 | 10.39 |  |
| £40,000 - £59,000 | 14.39 |  |
| £60,000 - £79,000 | 17.38 |  |
| £80,000 - £99,000 | 13.59 |  |
| £100,000 - £149,000 | 16.68 |  |
| >£149,999 | 10.39 |  |
| **Parental Education** | *Mother* | *Father* |
| Primary | 0.30 | 1.50 |
| Secondary | 10.60 | 22.40 |
| Tertiary | 88.90 | 74.90 |
| Unknown | 0.00 | 0.80 |

Note: Those that skipped each question have not been included in the percentage calculation for that question. For each category, the remaining percentage is for those who responded, "prefer not to answer".

## Viability

The type of technological approach is an important factor than can influence the viability of remote in-home monitoring. Our analysis is divided into the following subheadings as per the research topic: Sticker sensory electrodes versus smart suits, cot versus body image-only video recording and smartphones.

**Sticker sensing electrodes versus smart suits.** To examine acceptability and practicality, for each question we report comparisons between ratings assigned to sticker sensing electrodes and smart suits (for further analysis with comparisons to wrist-/ankle-bands see S5 in S2 File).

*Comparison of likelihood.* More participants indicated they would be more likely to use smart suits ($Mdn$ = 5.00, $M$ = 4.37, $SD$ = 0.79) than sensing electrode stickers ($Mdn$ = 3.00, $M$ = 2.91, $SD$ = 1.32), (see Fig 1A). Wilcoxon Matched-Pairs Signed-Ranks Test (WMPSR) revealed significantly more participants (72.56%) rated using smart suits more likely than sensing electrode stickers than the opposite (2.37%), $z$ = -14.24, $p$ < .001, with moderate effect size ($r$ = -0.52). Of the total sample, 24.80% rated participation with each technology as equally likely.

*Comparison of practicality.* Averaged across both lengths of time, 73.97% of participants indicated smart suits as more practical than sensing electrode stickers (Fig 1B). A Friedman Analysis of Variance (FANOVA) identified a significant difference in practicality rating between using smart suits for a "Short Period" ($Mdn$ = 4.00, $M$ = 3.50, $SD$ = 0.62), smart suits for an "Extended Period" ($Mdn$ = 3.00, $M$ = 2.85, $SD$ = 0.88), sensing electrode stickers for a "Short Period" ($Mdn$ = 2.00, $M$ = 2.14, $SD$ = 0.92) and sensing electrode stickers for an

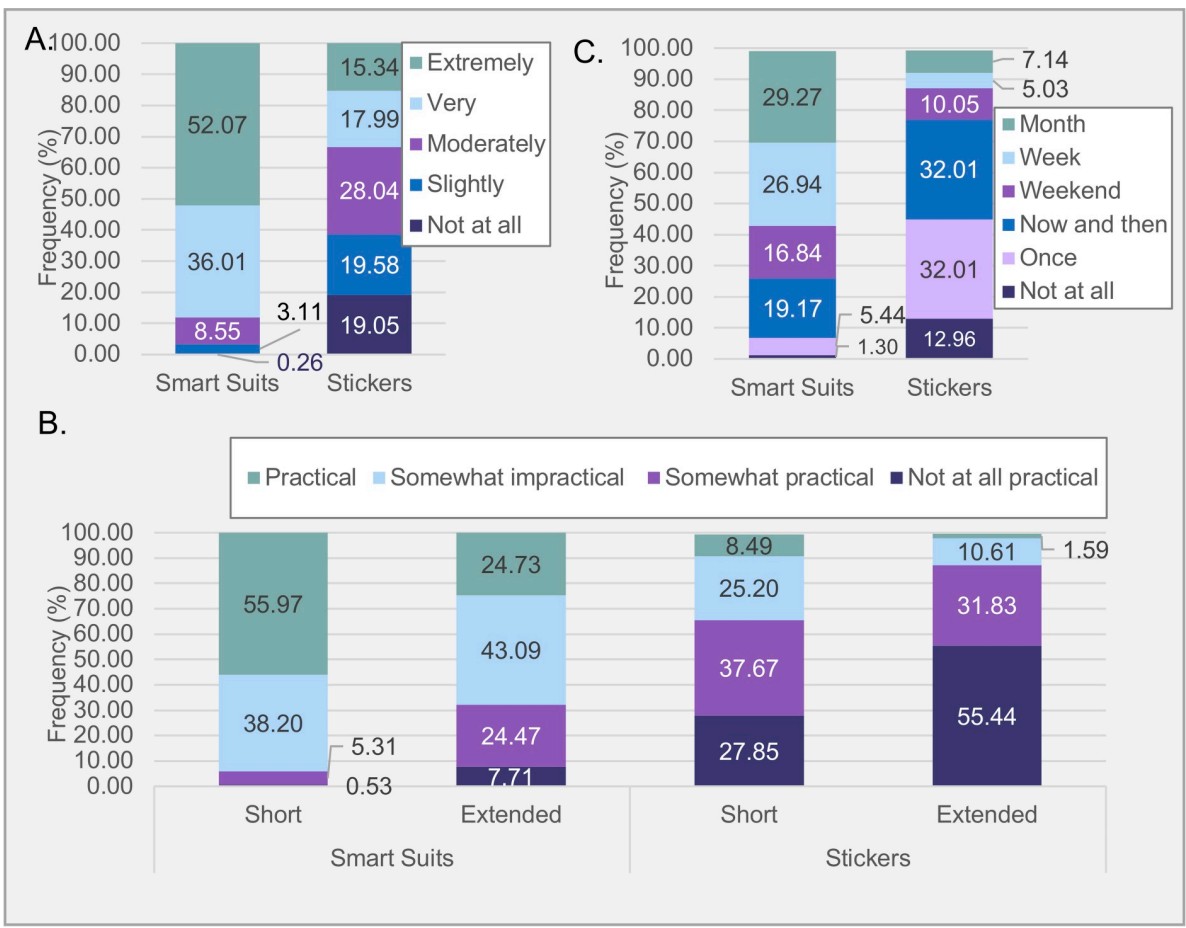

**Fig 1. Percentages of response frequency for questions concerning viability of smart suits and sticker (electrode) technologies. (A)** Participation likelihood rating; **(B)** Practicality rating over differing durations. **(C)** Preferred participation duration. *Note.* Percentages were calculated out of all responders who answered per question, including "Prefer not to answer" responses.

"Extended Period" ($Mdn = 1.00$, $M = 1.58$, $SD = 0.74$), $\chi^2(3, N = 358) = 686.17$, p < 0.001. A Kendall W of .639 was reported, indicating a good level of agreement among responders. Dunn post-hoc tests with Bonferroni adjustments (S7 Table 1 in S2 File) revealed responders significantly rated using smart suits for a "Short Period" as most practical ($p < 0.001$).

*Comparison of length of participation.* Participants were willing to use smart suits for extended periods ("Week" or "Month") but sensing electrode stickers were rated more preferred for shorter periods (Fig 1C). A Bhapkar test confirmed that participation length was significantly longer for smart suit than sensing electrode sticker technologies, $\chi^2(5, N = 372) = 554$, $p < 0.001$. See S7 Note A for detailed description of results in S2 File.

*Open-ended responses.* Concept maps depicting the prevalence and co-occurrence of themes and concepts in the open-ended response question for smart suits and sensing electrode stickers were generated (see Fig 2A & 2B). "Hygiene" was the most commented theme for smart suits. Concerns surrounding "Wires" was most commonly mentioned for sensing electrode stickers (see Tables in Fig 2A & 2B).

**Static versus body image-only video recording devices.** To examine acceptability of video recording with differing levels of active care, we compared participants ratings on static cot camera versus body camera for image-only video recording.

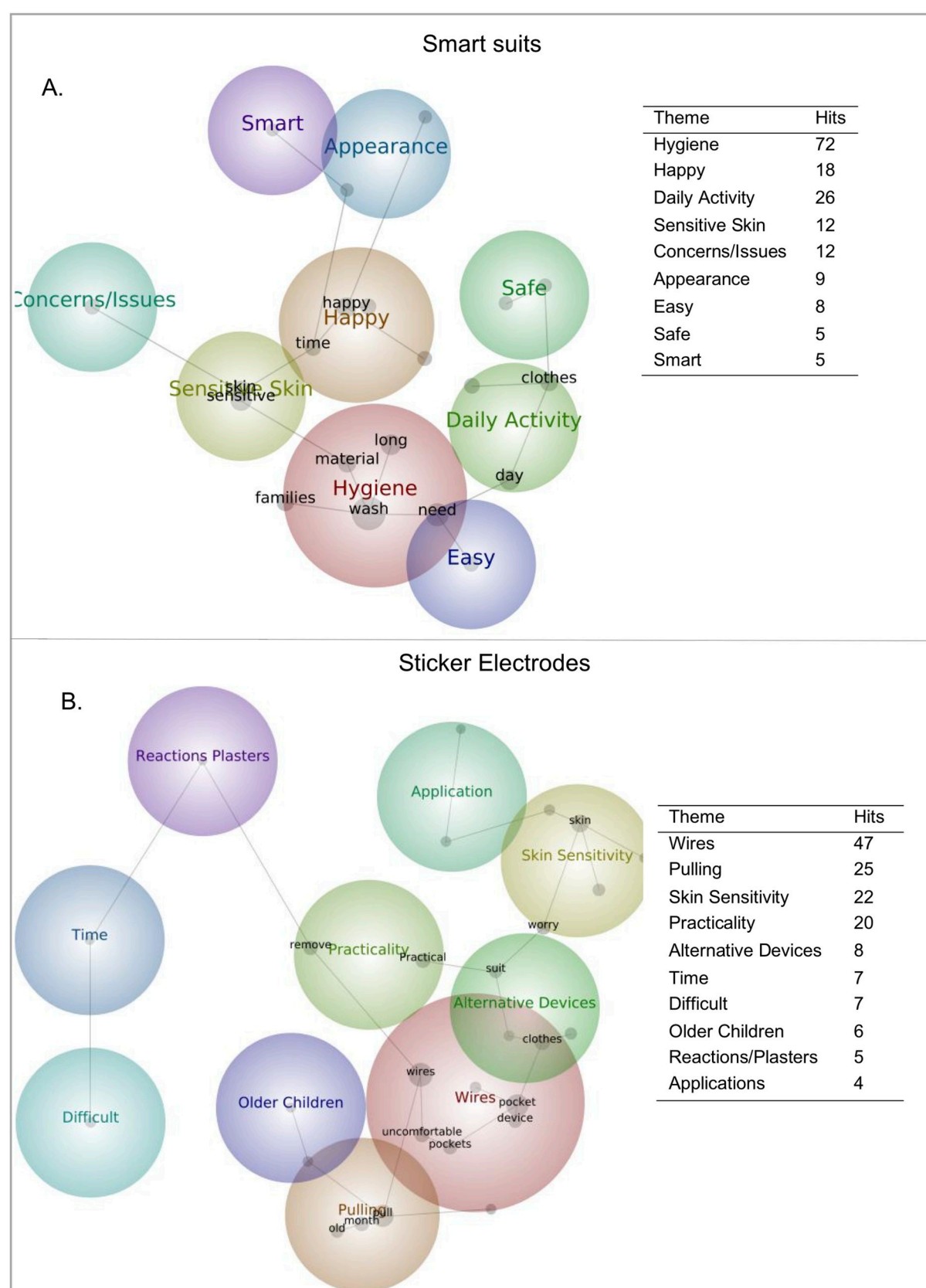

**Fig 2. Theme/concept maps depicting prevalence and co-occurrence of words from open-ended response questions for.** (A) smart suits and (B) sticker electrodes. Tables display each theme and its relative hit number for relative technology.

*Comparison of likelihood.* The majority of the group (76.31%) indicted no differences in likelihood ratings between cot camera (*Mdn* = 4.00, *M* = 3.79, *SD* = 1.18) and body camera (*Mdn* = 4.00, *M* = 3.79, *SD* = 1.16), (Fig 6A). Of the 23.69% of participants who changed their response, WMPSR revealed no significant preference between on-body camera and the cot camera, *z* = -112.00, *p* = .91.

**Smartphones.** *Comparison of likelihood.* Of the total sample, 91.37% of participants indicated they would be "Moderately" to "Extremely" likely to use a smartphone to record/interact with the study (Fig 3A). Median likelihood rating (*Mdn* = 5.00, *M* = 4.16, *SD* = 1.06) was

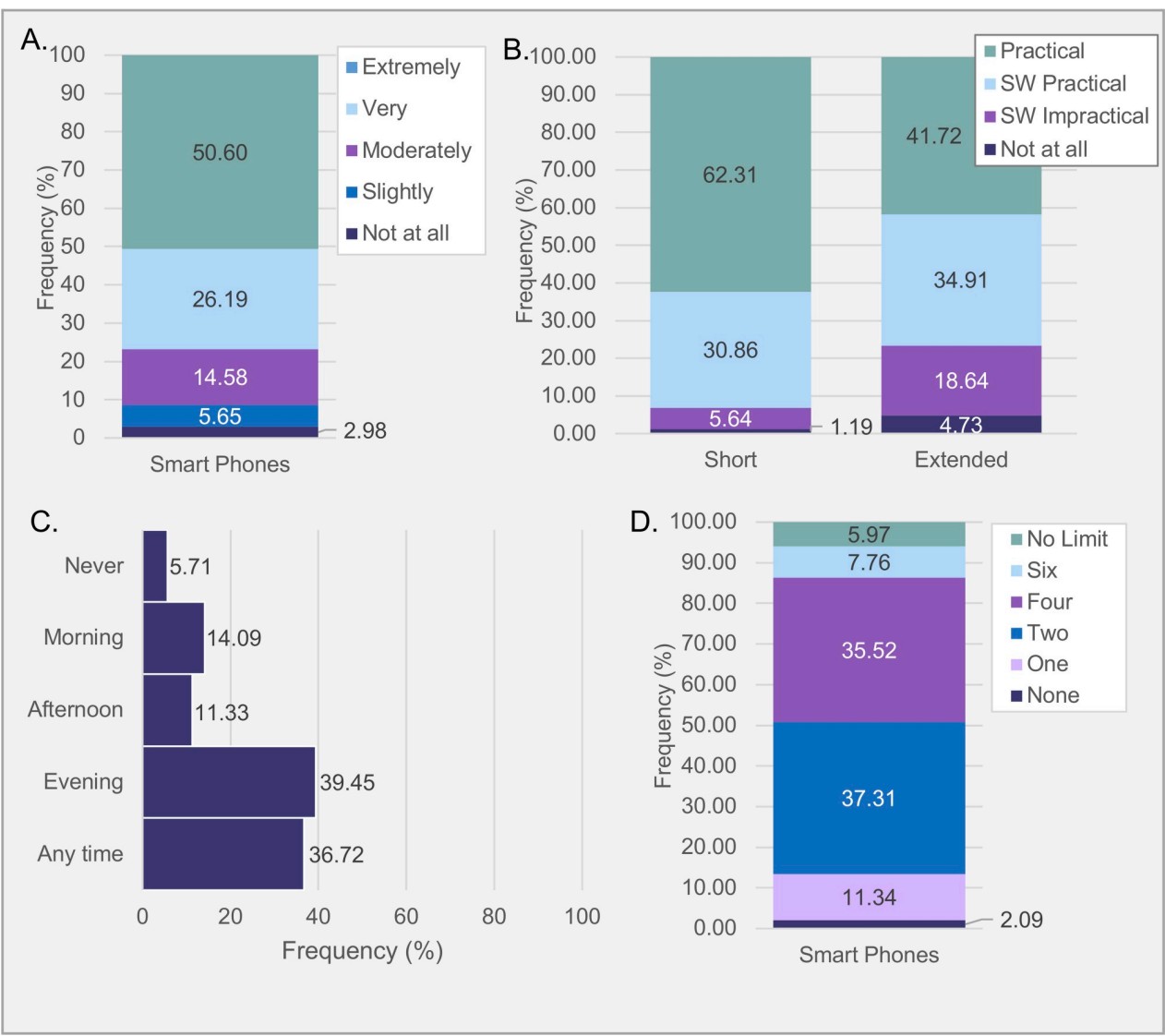

**Fig 3. Percentages of responses to questions on interacting with home testing via smartphones.** (A) Participation likelihood rating. (B) Practicality of different participation durations. (C) Best for immediate response. (D) Preferred prompt number. *Note.* Percentages were calculated out of all responders who answered per question, including "Prefer not to answer" responses. *SW = Somewhat.*

significantly higher than the score median (2.5), $Z = 15.169$, $p < .001$, with a moderate effect size ($r = .83$).

*Comparison of practicality.* Of the total sample, 62.02% of people did not think duration of the study would affect the practicality of smartphone use (Fig 3B). Overall, median practicality responses were equal for using the smartphones for a "Short Period" ($Mdn = 4.00$, $M = 3.11$, $SD = 0.81$) and for an "Extended Period" ($Mdn = 4.00$, $M = 2.63$, $SD = 0.89$). For the 37.98% of the sample who expressed a preference, WMPSR revealed significantly more participants (92.97%) indicated using smartphones for a "Short Period" more practical than over an "Extended Period" than those who indicated the opposite (7.03%), $z = -9.341$, $p < .001$, to a small effect ($r = 0.35$).

*Preferential response time.* Of the total sample, 73.33% of caregivers would be more likely to respond to smartphone prompts "Anytime" or in the "Evening" respectively, (Fig 3C). Proportions of preferred response time were unequal, $\chi^2(4, N = 335) = 155.49$, $p < .001$, with a small chance-corrected effect size (R = 0.11). "Evening" and "Anytime" were deemed more favourable than all other options (p < 0.001), with no significant difference between the two (*p* = 1.00; S8 Table 1 in S2 File).

*Prompt number.* Of the total sample, 77.46% of caregivers would be most likely to respond to 2–4 prompts a day (Fig 3D). Responses were not equally distributed, $\chi^2(5, N = 335) = 244.49$, $p < .001$, *Cramer's V* = 0.38. Bonferroni corrected binomial tests (S8 Table 2 in S2 File) revealed two prompts (39.68%) and four prompts (37.78%) were equally favoured (*p* = 1.00), and were both significantly more preferred than all other prompt options (*p* < 0.001).

*Open-ended responses.* Concept maps depicting the prevalence and co-occurrence of themes and concepts in the open-ended response question for smartphones were generated (see Fig 4). As indicated by the number of hits, "Time-consuming" was the most commented theme (see Fig 4).

## Privacy

In-home remote infant monitoring studies, particularly those with highly identifiable image-only video and audio recording, raise privacy issues. One way to address this concern is to pre-processing video and audio data such that the original identifiable recordings are not stored. We aimed to examine caregiver opinions according to the following two subsections: video and audio recording with/without privacy preservation and data sharing.

**Video and audio recording with/without privacy preservation.**  *Comparison of likelihood.* Respectively, 51.66%, 55.37% and 55.75% of caregivers did not consider privacy preserving technique to change their likelihood of participation with on-body camera, cot camera and audio recording. Of those who changed their response, 84%, 84.57%, and 86.67% preferred automatic processing options over manual processing, respectively for on-body camera, cot camera and audio recording (Fig 5A). FANOVA identified an overall significant difference in ratings of likelihood in accepting manually processed body camera ($Mdn = 3.00$, $M = 3.30$, $SD = 1.26$), automatically processed body camera ($Mdn = 4.00$, $M = 3.79$, $SD = 1.16$), manually processed cot camera ($Mdn = 4.00$, $M = 3.39$, $SD = 1.28$), automatically processed cot camera ($Mdn = 4.00$, $M = 3.79$, $SD = 1.18$), manually processed audio ($Mdn = 3.00$, $M = 3.43$, $SD = 1.17$) and automatically processed audio ($Mdn = 4.00$, $M = 3.85$, $SD = 1.11$), $\chi^2(5, N = 335) = 219.633$, $p < 0.001$. Kendall W of .131 indicating a weak level of agreement among responders. Dunn post-hoc tests with Bonferroni adjustments (S9 Table 1 in S2 File) revealed significant differences in ratings between manual and automatic processing for all technologies (*p* < 0.001), with automatic processing rated higher than manual. No significant differences were found within processing types between video and audio recordings (*p* = 1.00).

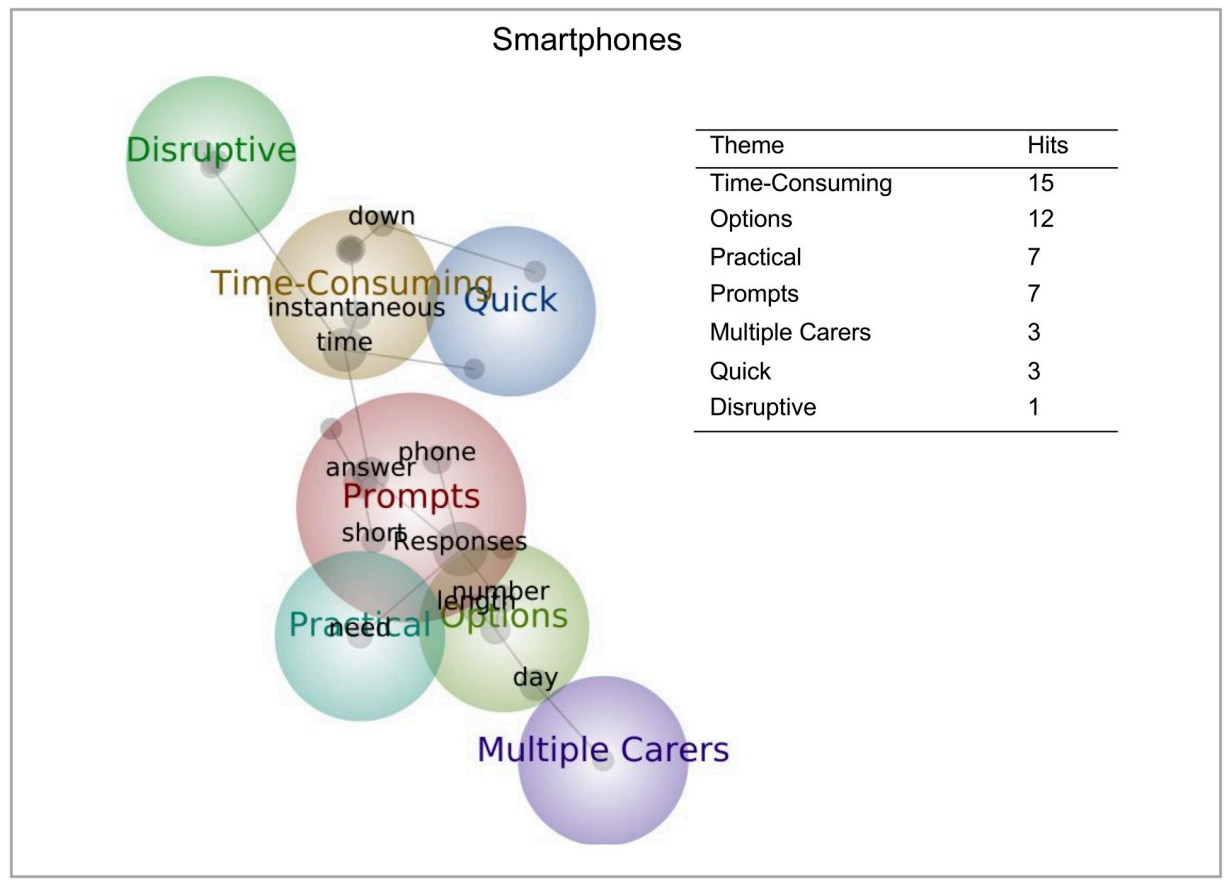

**Fig 4. Theme/concept maps depicting prevalence and co-occurrence of words from open-ended response questions regarding smartphones.** Table displaying each theme and its relative hit number.

*Comparison of practicality.* Averaged across both lengths of time, 58.31% of participants indicated video and audio as equally practical (Fig 5B). A FANOVA with pairwise Dunn post-hoc tests with Bonferroni corrections indicated a "Short Period" of video (*Mdn* = 3.00, *M* = 3.11, *SD* = 0.82) and audio (*Mdn* = 3.00, *M* = 3.11, *SD* = 0.81) more practical than "Extended Period" of video (*Mdn* = 3.00, *M* = 2.57, *SD* = 0.90) or audio (*Mdn* = 3.00, *M* = 2.63, *SD* = 0.89), $\chi^2$(3, N = 337) = 229.20, *p* < 0.001, (see Fig 5B). Though this was to a weak effect (Kendall W = 0.227), with 47.21% and 42.40% of the sample indicating "Short Period" of recording more practical than "Extended Period" for video and audio respectively. No significant difference to the ratings of practicality between video and audio recording for each recording session length (i.e. "Short Period" or "Extended Period") was reported (*p* = 1.00; S9 Table 2 in S2 File).

*Comparison of preferred length of participation.* Most caregivers favoured participation time "Now and Then" for all technologies except automatically processed video, for which most indicated they would be happy to use for at least a "Month". Averaged across recording technology, 66.67% of responders did not change their response when given the privacy preserving option. For the third of the sample that did change their response, 79% and 87.74% of participants opted for longer periods of privacy-preserved (i.e. automatically processed) video and audio recording, respectively, over their manually processed counterpart. Bhapkar tests revealed privacy-preserving processing significantly increased the length of time participants

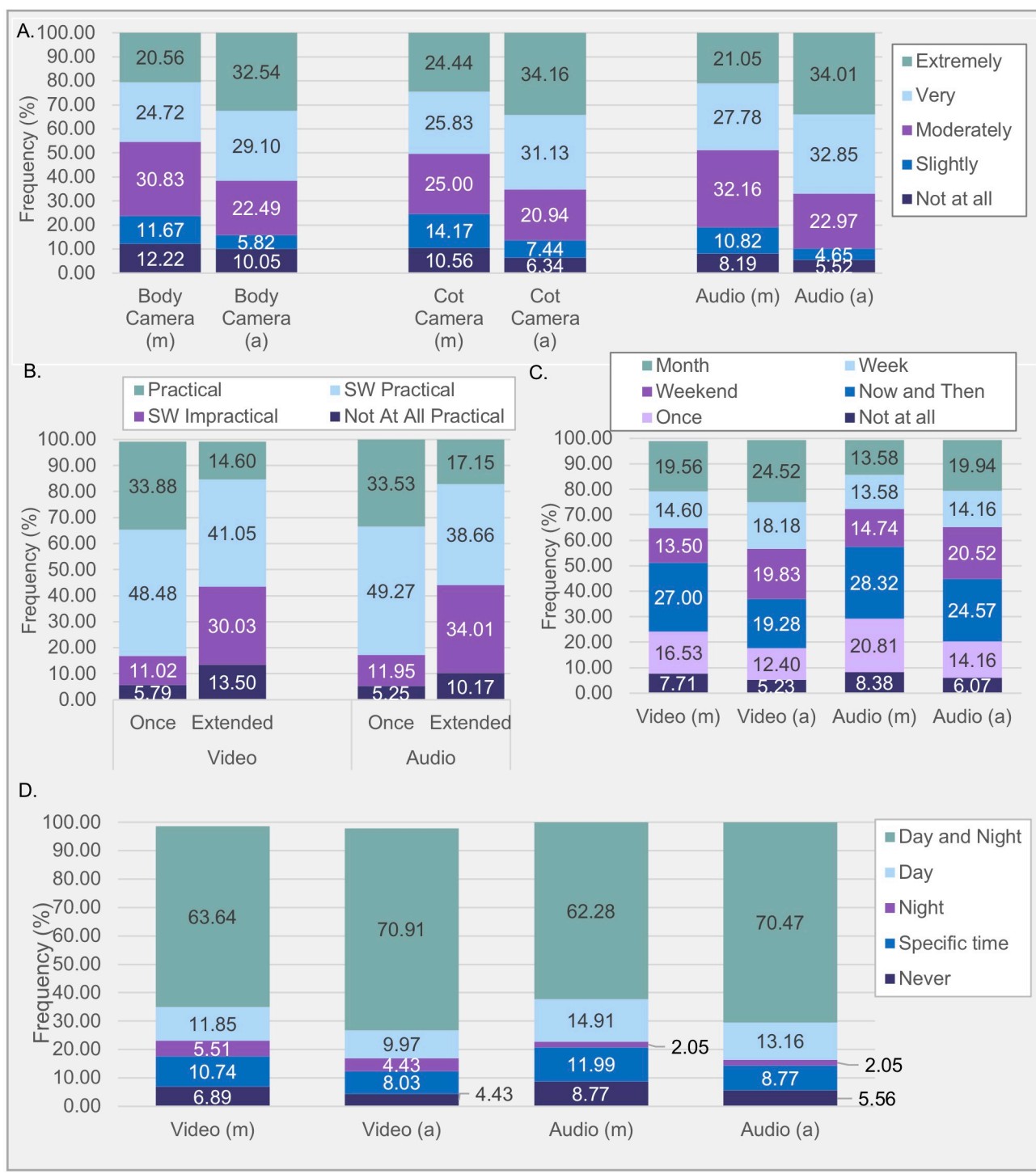

**Fig 5. Percentages of responses for questions concerning recording technology combinations within the theme of privacy. (A)** Participation likelihood rating. **(B)** Practicality ratings. **(C)** Preferred participation duration. **(D)** Preferred participation time. Key: (m) = manual processing, (a) = automatic processing. *Note*. Percentages were calculated out of all responders who answered per question, including "Prefer not to answer" responses *SW = Somewhat*.

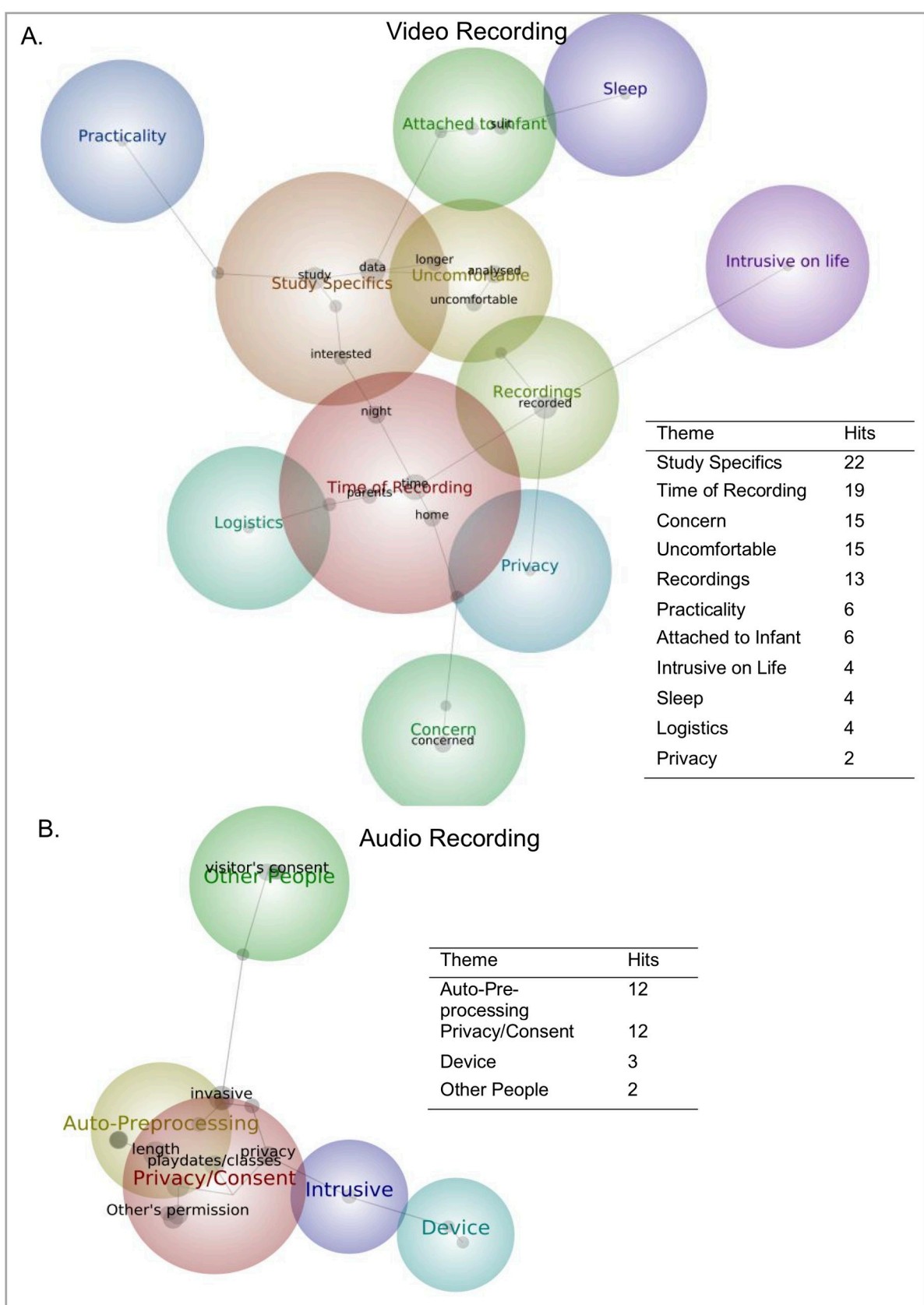

**Fig 6. Theme/concept maps depicting prevalence and co-occurrence of words from open-ended response questions.** (A) Video recording and (B) audio recording. Tables display each theme and its relative hit number.

were willing to record with both video, $\chi^2(5, N = 358) = 59.40$, $p < .001$, and audio, $\chi^2(5, N = 343) = 56$, $p < .001$, though this was to a small effect as displayed in Fig 5C. See S9 Note A for detailed breakdown of results on preferred length of participation for video and audio recording in S2 File.

*Comparison of when willing to participate per day.* The majority of caregivers were happy to participate in audio and video recordings during both "Day and Night" (Fig 5D). See S9 Note B for detailed breakdown of results on preferences on when caregivers were willing to participate per day in S2 File.

*Open-ended Response.* Concept maps depicting the prevalence and co-occurrence of themes and concepts in the open-ended response question for video and audio recording were generated (see Fig 6A & 6B, respectively). As indicated by the number of hits, comments on "Study Specifics" was the top theme for camera recording, whereas the themes of "Auto-Pre-processing" and "Privacy/Consent" were dually most commented on for audio recording (see tables in Fig 6A & 6B).

**Data sharing.** Of the total sample, 43.26% of caregivers were happier to share data more broadly when it could be anonymised (Fig 7A). For anonymous data, preferred research team was not equally distributed to a large effect, $\chi^2(4, N = 324) = 524.94$, $p < .001$, *Cramer's V =* 0.74; significantly more people said they would be most likely to consent to sharing data with "International" research teams ($p < 0.001$; S10 Table 1 in S2 File). Participants also indicated a preference of research team with whom to share non-anonymous data, $\chi^2 (4, N = 327) = 286.90$, $p < .001$, *Cramer's V =* 0.47; Participants equally preferred "BabyLab" ($N = 151$) and "International" ($N = 127$; $p = 1.00$) significantly above all other response options ($p < 0.001$; S10 Table 2 in S2 File). Bhapkar test indicated a significant effect of anonymity on data sharing preference, $\chi^2(4, N = 319) = 249.90$, $p < .001$. Of the 46% of responders who changed their response between anonymised and non-anonymised data, 100% chose more local research groups if their data was not anonymised (see S10 Table 3 in S2 File for McNemar post-hoc comparisons).

## Data access

Caregivers generally were interested in viewing their infant's data, with the highest interest in sleep (Fig 8B). Cochran's Q test indicated significant differences to the proportions of those who opted a preference to view each type of data, $\chi^2(5, N = 331) = 258.727$, $p < .001$, with a small chance-corrected effect size (R = 0.09); caregivers were more interested in viewing sleep data (90.90% of sample) than all other choices ($p < 0.001$), except for video (84.90%; $p = 0.56$) (S11 Table 1 in S2 File). The proportion of those who opted a preference to censor each type of data significantly differed, $\chi^2(6, N = 331) = 881.468$, $p < .001$, with a small chance-corrected effect size (R = 0.34); caregivers were more likely to want to censor video (80.70%) and audio data (75.50%) than all other choices ($p < 0.001$), with no significant differences indicated between these two technologies ($p = 1.00$; S11 Table 2 in S2 File).

## Future participation

Participants were very open to the prospect of participating in future studies similar to those described in the survey (Fig 7C & 7D). Of the total sample, 59% of responders indicating they would be "Very" or "Extremely" interested to in future investigations, with 57% of responders indicating they were eligible (had a child under 12 months) and were willing to participate.

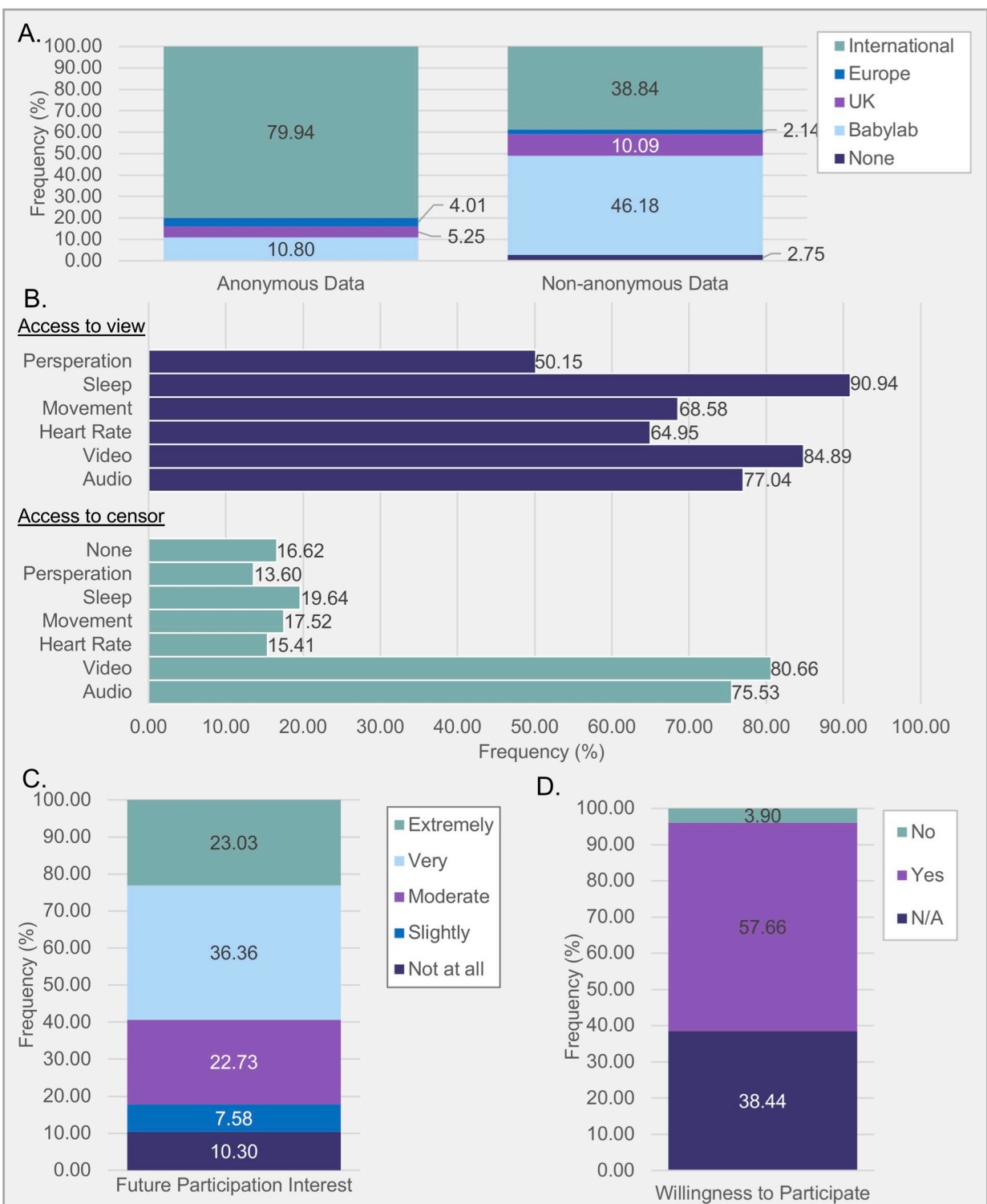

**Fig 7. Percentages of responses for questions concerning.** (**A**) Preferred data Sharing option; (**B**) Access to view and access to censor preferences; (**C**) Interest rating for future participation; (**D**) Willingness to participate.

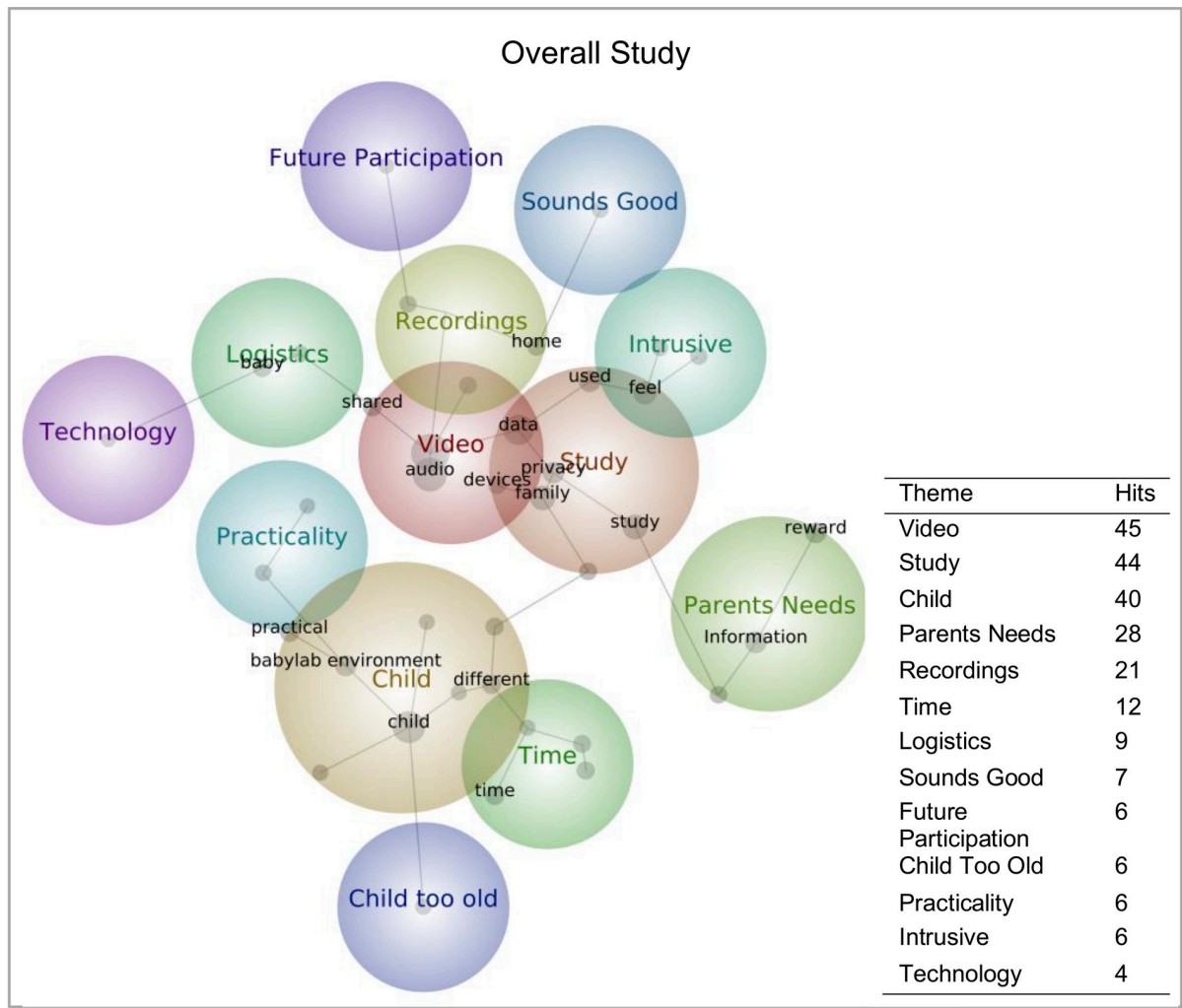

| Theme | Hits |
|---|---|
| Video | 45 |
| Study | 44 |
| Child | 40 |
| Parents Needs | 28 |
| Recordings | 21 |
| Time | 12 |
| Logistics | 9 |
| Sounds Good | 7 |
| Future Participation | 6 |
| Child Too Old | 6 |
| Practicality | 6 |
| Intrusive | 6 |
| Technology | 4 |

**Fig 8. Theme/concept map depicting prevalence and co-occurrence of words from open-ended response questions regarding the entire study.** Table displaying each theme and its relative hit number.

**Open-ended response.** Concept maps depicting the prevalence and co-occurrence of themes and concepts in the open-ended response question for the overall investigation are displayed in Fig 8A. According to the number of hits, the top three most commonly mentioned themes were "Video", "Study" and "Child". See table in Fig 8 for number of hits for all generated themes.

## Discussion

The present study investigated opinions of a UK based cohort of caregivers on remote in-home monitoring technology and study designs with an overarching goal to provide guidance for future research with similar aims. We discuss each of our core themes in turn.

### Viability

To maximise participant uptake and minimise attrition on studies implementing such promising technologies and assessment designs, we must optimise viability. To do so, we collected feedback on remote monitoring technologies such as wearable devices, video/audio recording and smartphone apps.

**Sticker sensing electrodes versus smart suits/sensing bands.** Participants rated smart suits (garments with integrated sensors) and sensing wrist/ankle bands device similarly, both being more favourable than the traditional stick-on electrodes commonly used in the lab and associated with generally better data quality. Smart suits and sensing bands were also rated more preferable for longer studies than sensing electrode stickers. Parents indicated a marginal likelihood of the longest duration of use for bands, possibly due to them being easy to use and thus least disruptive. Taken together, these results demonstrate the heightened acceptability of low-care remote monitoring technologies (e.g. smart suits and sensing bands), and allude to their potential of increasing uptake in remote monitoring study designs with infant subjects; particularly for designs taking place over longer periods. Notably, using smart suits for an "Extended Period" was indicated as significantly more practical than using sensing electrode stickers for a "Short Period", and participants rated a "Month" long usage as their most preferred option for sensing bands and smart suits, but not for sensing electrode stickers. This highlights the further potential of both smart suits and bands for longitudinal designs, with the possibility to decrease attrition in lengthy/time-intensive designs. This finding also warrants the investment in the development of sensing bands/suits for future studies (e.g. sensors wirelessly connected to apps that would pre-process data remotely, and upload anonymised derived data).

Concept/theme maps of the open-ended response items provide insight into concerns and reasonings behind corresponding multiple-choice answers (though it is important to note that these represent a simple summation of the most common words used, and future work should employ more extensive qualitative methods). The most commonly used words indicated a consensus across all three wearable technologies; that the devices must not impact on the child's physical health/wellbeing (e.g. "sensitive skin" for smart suits and sensing electrode stickers and "circulation" for bands). This finding was consistent with a previous qualitative report on potential barriers for participation with ambulatory infant sensing devices; caregivers expressed concerns about the comfort of the physical placement of the sensor on their child [28]. Though such concerns are likely to be addressed by manufacturers of the technologies as well as local ethics boards prior to implementation, future researchers should explicitly state the safety of these measures during advertising/consent to improve participant uptake.

For smart suits, the top theme was "Hygiene". This consisted of concepts "Wash", "Material" and "Nappy", highlighting a top concern for caregivers regarding smart suits was the cleanliness of the device. Reiterating this concern of hygiene, 76% of the sample indicated they were not willing to use smart suits other families have. For sensing bands, the top theme was "Water/Baby-Proof". This theme reflects concerns to the durability of equipment when being worn by the infant. For sticker electrodes, the top theme was "Wires", consisting of concepts "Wires", "Pocket", "Uncomfortable". "Wires" overlapped with other themes of "Pulling" and "Alternative Devices". Taken together, this highlights main topics raised by caregivers to centre around the relative safety, practicality and comfort of the sticker electrode wires compared to other devices. This disparity in the top extrapolated themes may account for the differing viability rating between these technologies. It is plausible that smart suits and sensing bands were rated more favourable than sticker electrodes because concerns of safety and comfort issues outweigh issues of hygiene and durability of the device.

It must be noted that although bands/body suits were rated more favourably in the quantitative rating, stickers were not completely disregarded by caregivers. Potentially, by addressing the highlighted concerns, all three monitoring technologies may be seen more favourably in future research.

**Static versus body image-only video recording devices.** Caregivers indicated similar levels of usage likelihood for static cot cameras versus on-body camera for image-only video

recording. This results indicates that for the collection of a highly identifiable video recording, the means with which this data is collected is not of any importance. The implication of this finding for future researchers is that studies involving video data can be flexible with their recording approach.

**Smartphones.**   Caregivers were generally positive about smartphones in the context of home recording studies. A majority of responders indicated that they would be "extremely" or "very" likely to use this approach, which is consistent with the increasing number of publications using this design for self-monitoring of psychological and behaviour states [36]. Both short or long periods of smartphone use was rated above the median practicality; the majority of responders rating both recording lengths to be equally practical. This finding aligns with the ubiquity of smartphones in society and the notion that using smartphones for research interaction are unlikely to cause significant burden on participants [8]. This overall favourable rating of smartphone use likelihood and practicality, as well as the additional benefits of smartphones (e.g. sensing device interaction-platform; [37]), highlight a positive prospective for the inclusion of smartphones in future remote infant investigations.

An additional benefit of using smartphones in remote investigations is the timely collection of additional data (e.g. questionnaires) through prompts/notifications. The majority of participants would prefer 2–4 prompts per day, which if implemented would allow multiple data collections or allow large data collections to be spread across multiple time points. However, as indicated by two time-related themes on the theme/concept map "Quick" and "Time-consuming", caregivers raised that responses need to be short, particularly for those requiring "immediate" responses. If multiple prompts/collection times are desired, future investigations should consider responses to be short and quick to minimise data loss.

A relatively large proportion of responders were happy to respond "Anytime" to prompts. Having immediate responses throughout a 24-hour period would reduce retrospective data recording, that may impede validity and increase the possibility for skewed results to particular times in the day/behaviour. This would defer from the ubiquitous benefits of remote monitoring that is often unachievable in lab-based study designs [8]. The majority of those who indicated a preference indicated "Evening", highlighting how responses are likely to covary with less busier times of the day (e.g. in the evening when the infant is likely to be asleep). The notion of differing schedules was raised within the theme of "Options" in the open responses. Many comments in this theme centred around the variability of daily life, and how this could determine the ability to answer prompts; thus, needing response options pertaining to the "number", "day" and "length". Implementing a means with which participants can easily communicate with investigators (e.g. smartphone app) about daily/weekly availability will enable researchers to establish whether data is missing at random or whether it is related to other factors (e.g. busier family); an important consideration to prevent incorrect conclusions.

Similarly, family dynamics can fluctuate both across families and within families across time. This was highlighted as a key consideration for caregivers by the "Multiple carer" theme. When requiring multiple data collection points or employing a longitudinal design, investigators need to consider how they may overcome reporting consistence when a child has different caregivers across different times of the day/week. Facilitating data collection across multiple caregivers (for example, using an app that can be downloaded on multiple devices with different family member/caregiver platforms) will allow families with complex structures to participate.

Ultimately, accounting for individual schedules will be important for future remote studies in general, particularly when employing experience sampling methods.

## Privacy

In-home infant monitoring studies can be somewhat intrusive on privacy. To optimise the success of such study designs, we must consider caregiver's feedback on different types of highly identifiable data collection and possible privacy-preserving techniques as well as with whom participants are comfortable sharing remotely collected data.

**Image-only video versus audio recording with/without privacy preservation.** Similar to smartphones, audio or image-only video recording not only contribute contextualising data, but additional informative data–thus making them a promising, and increasingly used tool in infant remote monitoring studies [11,45–47]. There was no effect of recording technology (i.e. video/audio) on likelihood ratings. Privacy preserving techniques (i.e. data anonymised on collection via automatic processing) did increase favourability of technology but to a moderate degrees, with approximately half of participants responding the same with and without the option of privacy preserving processing within technology. We also considered opinions on practicality of video and audio devices, as well as most preferred time of day to record with each recording device/processing option combination. No significant differences between technology were reported and there was little difference between device/processing combination preference for time of recording, with "Day and Night" significantly the most preferred for all combinations. Taken together our results are consistent with previous findings of no difference in willingness between video and audio recording and is somewhat consistent with previous non-significant trends to increased recording willingness if privacy-preserving techniques are applied [28].

The lack of preference to either image-only video or audio, as well as the moderate favourability to privacy preserving processing was reiterated in responses to preferred study duration. The majority of participants did not change their response between technologies nor between privacy preserving option. Of the sample that did change their response within each technology, privacy preserving processing option significantly increased preferred study duration. Although this was only for approximately a third of the sample, this finding still highlights potential of privacy preservation to increase uptake in longer studies. The level of privacy and intrusion when conducting remote infant monitoring studies with identifiable data should be considered by future investigators, particularly for longer studies. This preference to privacy preservation corroborated in the open-ended response sections whereby theme/concept maps extrapolated "Privacy" and "Intrusive" as themes for both video/audio recording, and "Auto-Pre-Processing" for audio recording. Yet, implementing such measures for uptake improvement can be problematic. Anonymising data on collection means raw data will not be stored. Researchers must therefore thoroughly consider beforehand what variables are needed for analysis, and the validity of extracted data using the chosen method given the data collection setting (e.g. will the infant's vocal pitch be validly extracted if collected in a noisy household). Although this fits with pushes towards preregistration of analytic plans [48]; it limits opportunity for prospective secondary analysis, particularly when raw data is no longer available. This reduces scope for cost effective and efficient research where unforeseen questions that have arisen from the results of the initial analysis can be answered with the original dataset. Future researchers must consider whether the potential increase in uptake/decrease in attrition outweighs future secondary/unforeseen analysis limitations, particularly when conducting time and resource intensive studies.

In the open-ended responses to audio recording, the themes of "Privacy/Consent" and "Other People" were extracted interconnectedly, encapsulating raised concerns about recording people without consent (e.g. visitors). The issue of recording others was also raised in the video open-ended response (particularly centred around issues with nursery), though was not

prevalent enough to warrant its own theme or concept. These concerns raise the important logistical consideration for future researchers to align with General Data Protection Regulations [38] by taking the consent of all participants, including those who are not necessarily enrolled in the study but are nonetheless in recordings (e.g. visitors). This is problematic for busy families; whose infant may have multiple caregivers and/or participate in multiple activities where there are other people. A number of caregivers indicated recording would be problematic as their infant goes to nursery where video/audio recording is prohibited. Indeed, recording could be limited to in-home when only the infant's family is present–but such specific circumstances may limit the range of data collection, potentially skewing results away from all-encompassing naturalistic observations. Alternatively, recording only in the home may dissuade more active families who spend a lot of time out of the home–a concern raised by caregivers and mentioned within audio's "Other People" theme and "Playdate/Classes" concept and video's "Home" concept. Studies requiring long periods within the home may not be accommodating to every family daily life. If these families were to stay indoors, recording would no longer be completely naturalistic. Alternatively, such families may not participate, creating generalisability issues. This is an important issue that needs careful consideration by future investigators.

Our findings ultimately suggest an initial high and equal acceptance of video and audio recording, which is slightly increased when privacy persevering techniques such as automated processing are applied. However, acceptability may differ when such studies are proposed in the context of more lengthy designs, where future investigators must consider the type of recording device and the application of privacy preserving techniques alongside more complex issues such as consent of those who are not immediately involve in the study.

**Data sharing.** Findings from our question on data sharing uncovered an effect of anonymity on data sharing preference. The majority were willing to share anonymous data with "international" research groups. For non-anonymous opinions were split, "BabyLab" indicated only minorly more favoured to "International" research groups. This suggests there may be two subgroups within our sample that have differing thoughts on privacy with regards to data sharing, a consideration that we preliminarily explored (see S14 Note A for analysis in S2 File). Maternal education was associated with the preferred choice of research group with whom to share non-anonymous data. These findings indicated that maternal education may influence readiness to share data internationally, a notion that should be investigated in future research.

Taken together these results somewhat corroborate the above findings highlighting a preference for anonymous data. Although anonymising data may provide a route to appease privacy concerns, it may be problematic when considering pushes for the Open Data Movement [40]. Having participants unwilling to allow their data, particularly raw non-anonymous data, to be used outside of the recruiting research group may limit transparency and reproducibility (thus potentially the credibility) of findings, and impede research collaborations [39,40].

## Data access

We wanted to gauge whether caregivers wanted access to their infant's data to view or to censor (i.e. OK/Delete collected data). The majority of participants were interested in viewing sleep, video and audio data. Given the interest in viewing data, plus previous reports of data access increasing willingness to participate [28], it could be considered a priority for researchers to incorporate a data access option to encourage enrolment onto such intensive investigations. This need for encouragement is backed by the "Reward" concept generated in the theme of "Parents' Needs" from the open-ended responses regarding the overall investigations. To illustrate, one commenter said "sounds like a big commitment from parents. It would need a

reward". For censoring purposes, the strongest preference was for video and audio recordings. This highlights caregivers' wish for an added layer of privacy on identifiable data by being able to select what is given to researchers. Although providing this option may increase uptake, it has the potential to skew results to only more flattering views of the infant's life—limiting obtaining comprehensiveness. Future researchers need to carefully consider the purpose of data access and hence ensure data access is informative and relevant for caregivers, without being overwhelming, complex or concerning.

## Future participation

Over half the sample indicated they would be "Extremely" or "Very" likely to participate in future remote in-home monitoring investigations, and a similar amount said they had an eligible infant (younger than 12 months; S12 Table 1, 2 in S2 File). Taken together, these findings are promising, suggesting a high level of acceptability for such investigations. Many open-ended responses corroborated this notion, as highlighted by the generated themes of "Sounds Good". Other themes and concepts reiterated previously mentioned concerns, particularly regarding "Technology", "Privacy" issues of "Video" and "Audio" recording, "Intrusive[ness]" and "Practicality". Additional topics previously unmentioned were highlighted. The concept of "Babylab Environment" was generated, which referred to how the babylab environment made caregivers more confident with the equipment that was being deployed on their child. Taking similar equipment out of the babylab may require a lot more briefing on the technology, how it works and how to implement it–which may be a big commitment for time-restrained participants (as suggested in the theme of "Time"). Therefore, future researchers must carefully consider how and to what level is necessary when informing participants.

## Limitations

A major limitation of the survey is that our sample may not have been representative. Firstly, the mean of our target child was 29 months. This means that some of the caregivers were answering the questions retrospectively, which may have altered their opinion. Similarly, our sample may not have represented families that would benefit most from in-home remote monitoring technologies. Families of low SES tend to be underreported in lab-based investigations as, among many reasons, travelling to the lab is a burden on limited resources [4]. We used Birkbeck University of London's Babylab database and social media accounts to recruit participant, thus people who responded to the survey were probably those already likely to engage in and would have an already high level of acceptance of our research, limiting the scope of generalisation to a degree. The recruitment stream used may have particularly limited access to low SES families, as noticed in the demographic of our sample. This illustrates that to engage boarder communities targeted outreach efforts, alongside the proposed accessible in-home study designs, need to be addressed in future investigations.

In light of this limitation we conducted additional analysis on a subset of individuals from our sample who were of relatively lower SES (e.g. infant's caregivers' education was no higher than secondary and the family was in the lower third of annual income; S13, in S2 File). Findings from this small sub-sample (20 responders) were relatively similar to the overall sample's response. Trends demonstrated smart suits were preferred over sensing electrode stickers, smartphones were relatively accepted, and automatic processing of video/audio recording may increase acceptability. Similarly, automatic processing may increase willingness to share data internationally. However, this analysis is, of course, underpowered.

Several questions in our survey were open-ended response, requiring qualitative analysis to draw inferences on response themes/concepts. The motivation and benefit of collecting of this

data means topics of interest for caregivers, that may not have been considered priori, can be investigated. We attempted to conduct a rudimentary qualitative analysis using topographic maps that extrapolated themes/concepts based on the number of mentions. However, this was not an exhaustive systematic analysis of the qualitative data, therefore limits the thoroughness of conclusions we can draw.

When describing potential technology, we incorporated an example image in order to contextualise the question. Some of these images were of commercially available technology, which, are often more aesthetically pleasing than equipment that is developed in the lab without multidisciplinary collaboration. This may have inflated acceptance in our sample, and must be considered by future researchers who are using our findings to guide the implementation of less aesthetic technology. Similarly, the descriptions in the survey for the purpose of collecting each type of data was limited, in order to improve the generalisability of our findings. However, the description of the technology may have influenced responses, which may differ with more information or because of study specifics. Information pertaining to security and participant confidentiality for questions regarding data sharing were also limited. Providing information on the protocols of data across specific collaborating labs may influence opinions on data sharing. Ultimately, the level of participant understanding of the technology/assessment specific is likely to influence opinion, thus, needs to be taken into account by future investigators.

In order to comprehensively canvass caregivers' opinions on remote monitoring technologies and study designs, we asked a large number of question. Consequently, we conducted a large number of comparisons for which we attempted to correct for multiple comparisons within each analysis using Bonferroni correction. Nonetheless there is the potential for false positives. Therefore, we present all analysis alongside appropriate effects size and graphs, and have made the data available. This is to enable future investigators to evaluate the information that be most relevant to their particular study.

## Summary

The present survey indicates remote in-home infant monitoring to be perceived as feasible and acceptable by families with infants. The development of new technologies thus holds great potential for researchers to conduct naturalistic investigations remotely. This (if coupled with appropriate outreach approaches) may enable researchers to incorporate individuals who may not usually be able to participate in lab-based research e.g. families with more restricted resources. Although generalisability should be considered, our work indicates significant areas to be addressed to take this new frontier of infant research forward. We have summarised our top five key factors in Fig 9. Caregivers must feel confident that wearable equipment is not disruptive or impede on their safety. Caregivers raised important practical issues for remote designs regarding individual schedules and multiple caregivers. This could be overcome by participants interacting with the study using smartphone applications, which were considered as highly favourable. Privacy preserving techniques on video and audio data were deemed somewhat important, particularly when participating in more lengthy investigations. This requires researchers to carefully plan the variables they need to extract before the study begins, which limits raw data availability for replication and further secondary explorations. Reiterating privacy concerns, data access for censoring purposes on video and audio was rated highly, which may increase acceptability though may introduce biases with regards to what data is available for analysis. The tension between privacy issues and data sharing was also highlighted, with innovative new solutions required to comply both with legal frameworks governing data sharing, the views and wishes of caregivers to maintain their privacy and

Call Out Box of top 5 factors for future researchers to consider when conducting in-home monitoring with infant participants in the UK.

1. Remote technologies with reduced active care are more accepted by caregivers, thus investment to improve data quality of these technologies is justified.

2. Privacy persevering techniques for non-anonymous data recording modalities are likely to moderately increase the participation for longer study durations, but limit potential for future secondary/unforeseen analysis.

3. Though feedback is highly favoured, participant's motivation for feedback (e.g. curiosity or censoring) is likely to depend on the type of data and will influence the format of the feedback.

4. Researchers must consider participants preference for what and with whom data is shared, and thoroughly explain how their data will be handled when shared outside of the EU GDPR's governance.

5. For video and audio recording, researchers need to consider how they will take consent from friends and family who are not part of the infant's household but are likely to be captured in the recordings, e.g. visitors.

**Fig 9. Call out box of top five factors for future researchers to consider when conducting in-home monitoring with infant participants in the UK.**

geographical data sharing concerns (particularly for non-anonymous data), and the drive towards greater sharing of raw datasets. Taken together, our work highlights both opportunities and challenges associated with moving towards remote infant research practices.

## Supporting information

**S1 File. Survey.**
(PDF)

**S2 File. Analysis supplementary information.**
(DOCX)

## Acknowledgments

We would like to thank the participants who completed our survey as well as Dr. Lucas Noldus and team for helpful comments on the manuscript and support.

## Author Contributions

**Data curation:** Laurel A. Fish.

**Formal analysis:** Laurel A. Fish.

**Funding acquisition:** Emily J. H. Jones.

**Investigation:** Laurel A. Fish.

**Project administration:** Emily J. H. Jones.

**Supervision:** Emily J. H. Jones.

**Writing – original draft:** Laurel A. Fish.

**Writing – review & editing:** Emily J. H. Jones.

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
