## [Decision Letter · Decision Letter 0]

25 Aug 2020

PONE-D-20-20163

A survey on the attitudes of parents with young children on in-home monitoring technologies and study designs for infant research.

PLOS ONE

Dear Dr. Fish,

Thank you for submitting your manuscript to PLOS ONE. After careful consideration, we feel that it has merit but does not fully meet PLOS ONE’s publication criteria as it currently stands. Therefore, we invite you to submit a revised version of the manuscript that addresses the points raised during the review process.

We look forward to receiving your revised manuscript.

Kind regards,

Barbara Schouten

Academic Editor

PLOS ONE

Journal Requirements:

2. Please amend either the title on the online submission form (via Edit Submission) or the title in the manuscript so that they are identical.

3. Please include additional information regarding the survey or questionnaire used in the study and ensure that you have provided sufficient details that others could replicate the analyses. For instance, if you developed a questionnaire as part of this study and it is not under a copyright license more restrictive than CC-BY, please include a copy, in both the original language and English, as Supporting Information

4. Please note that PLOS ONE uses a single-blind peer review procedure. We would therefore be grateful if you could include in the information that has been redacted for peer review in the manuscript.

Reviewers' comments:

Reviewer's Responses to Questions

**Comments to the Author**

1. Is the manuscript technically sound, and do the data support the conclusions?

Reviewer #1: Partly

Reviewer #2: Yes

Reviewer #3: Yes

Reviewer #4: Partly

2. Has the statistical analysis been performed appropriately and rigorously? 

Reviewer #1: Yes

Reviewer #2: I Don't Know

Reviewer #3: Yes

Reviewer #4: Yes

3. Have the authors made all data underlying the findings in their manuscript fully available?

Reviewer #1: Yes

Reviewer #2: Yes

Reviewer #3: Yes

Reviewer #4: Yes

4. Is the manuscript presented in an intelligible fashion and written in standard English?

Reviewer #1: No

Reviewer #2: Yes

Reviewer #3: Yes

Reviewer #4: Yes

5. Review Comments to the Author

Reviewer #1: I will start by saying that I am aware that I am not supposed to be reviewing for "impact" or "interest level" and I have tried to bear this in mind in my review. I am not sure whether some of my comments have more to do with "clarity" or "impact", so I will present them here and leave it between the editor and the author how/to what extent they should be addressed.

Please note also that I am somewhat unfamiliar with some of the statistical tests used and with the qualitative analysis. However, to the extent that I am able to judge, the statistics appear appropriate and technical standards are met. I have no concerns about the quality of the research that was conducted, and the study is presented in a very transparent way.

My primary concern is the framing of the study appears to be centred around the needs of the particular lab for a study they are planning. However, they attempt to draw much broader conclusions that would be of interest to the larger infant research community, and the introduction sets up these themes in very broad terms. I don't believe that they have been entirely successful in drawing conclusions of relevance to the broader research community, and I don't know whether it would work to frame the study solely around the conclusions relevant to this particular lab, or even specifically labs that do smartsuit testing (it's not clear to me how many that is)?

I selected "no" for intelligibility not being there is a problem with the clarity of the English per se, but rather because there are some structural issues that make reading challenging. Two broad concerns: 1. The sub-headers are inconsistent, making it hard to match up the research questions with the data use to test these questions. For example, it took me a fair amount of searching to figure out what the sub-section "Study Interaction" was referring to. It appears to be a sub-section of the question(s) about practicality, but is listed separately in the Analysis section. 2. Relatedly, there are quite a lot of questions that are tested in a number of different ways, and the reader gets a little lost in the trees for the forest. I did not find the figures particularly helpful in this respect in visualizing the findings - the authors could take a step back and consider which aspects of the findngs they most want to highlight.

Throughout the study it is difficult to know the extent to which some of the findings might generalize beyond the specific lab/study under question. I went back to the wording of the questionnaire on a few occasions for this reason. A few things jumped out at me: 1. The described purpose of the audio/video may play a key role in the participants responses. Here the survey suggests that the need for video/audio is fairly limited so participants may be more inclined to request that the data be anonymized. In my lab, participants tend to be quite willing to share their raw audio with us, as they understand that it is necessary for the research. 2. The question about sharing with researchers outside of the lab does not describe any processes that might be in place to ensure participant confidentiality, which may also skew the responses toward a more conservative answer.

There are also a few typographical issues (e.g. "Lena" should be "LENA", "on-significant"?)

Reviewer #2: The authors conducted a survey with 410 parents on their perspectives on the challenges and ethical concerns of use of in-home infant monitoring technologies. The authors found that parents were more likely to accept video and audio recording in the home if data was anonymized and that they were open to the international sharing of the anonymized data. The authors generally found positive support for in-home monitoring devices and studies, but noted specific considerations surrounding privacy, safety, and family dynamics for different technologies.

This article is extremely relevant and critical for the field for the current pandemic and for the future of the field. Though many labs are quickly adapting to in-person testing restrictions, we know little about how families perceive the use of such technologies in their homes. I agree with the authors that a model that mitigates the burdens and sacrifices on the families will be essential to the growing adoption of research technologies by families.

I have some general suggestions for the authors, and more detailed notes below. First, I found the structure of the paper a bit confusing where in the Introduction and Research Questions there seem to be three main areas of focus (i.e., Practicality, Privacy, and Feedback), but the Analysis plan and Results separate those themes into a few more headers, and then the discussion creates new sub headers as well. My preference would be for more alignment or a way to introduce this organization to the reader. Because there is a lot of information in this manuscript, having some consistency in structure would be helpful for the reader. Second, I felt like the themes raised in the Introduction were at times broader than the scope of the article. For example, the authors discuss many issues related to practicality but it seems that this is examined specifically in relation to duration of use in a day. The authors also raise an important issue of feedback and the potential of noticing atypical behaviors, but these concerns do not seem to be addressed in the analyses and the discussion. I would suggest specifying the scope of the paper earlier in the introduction and research questions. Third, the authors present this work to be important for COVID-safe measures and ‘socially-distanced’ data, which of course is extremely relevant to our times now. However, adopting in-home infant monitoring technologies are likely important for issues of including many different families who may not always have the time or access to research studies, as well as performing research at a very high scale (as the authors acknowledge). I think it would benefit the manuscript to emphasize these latter points, because the implications of this work are much broader than for the pandemic alone. Finally, the authors seem to focus on specific comparisons between technologies in their analyses, which are not immediately obvious to the reader. It would be helpful to include this specificity in their research questions.

Introduction

- Page 4 line 69: “Although user-friendly, commercial devices allow limited access to raw, high resolution data and are therefore poorly equipped for more complex research questions” – I don’t understand why high-resolution data is poorly equipped for more complex research questions? It seems that high-resolution data would in fact be great for more complex research questions?

- The authors discuss practicality here in many ways, but the analyses seem to focus on duration of use in a day? I suggest to specify that focus here to help orient the reader. The terminology of practicality and practicability are also used interchangeably, I would suggest using one for consistency unless they convey different meanings.

- I’m not sure I understand the paragraph on the ‘contextualization’ of data (starting line 114). Aren’t all in-home technologies providing the contextualization of a more ecological setting?

- The authors mention the use of resolution in a couple of places but it doesn’t seem consistent (e.g., line 69 high resolution data vs. line 115 lower resolution data like physiological measures). Are physiological measures considered lower resolution because they are not identifiable like audio or video recordings?

- Rather than “Feedback”, perhaps using a different term specific to families’ access to the data is better? It doesn’t seem like the authors examine findings related to the issues of caregiver concerns of atypical behavior.

Method

- I appreciate that the authors included general comment boxes for open-ended responses.

- How were families compensated?

- Do the authors have information on how long it took caregivers to respond to the surveys (e.g., mean and range) ?

- I had difficulty understanding this sentence on page 12 lines 260 – 263: “Note that all McNemar tests in this paper conducted for each categorical option on 2*2 contingency tables constructed with number of responders selecting that categorical option versus number of responders not selecting that category across the items being assessed e.g. technology.”

- Is there a table that could help readers understand the different technologies targeted in the survey?

- “Now and Then” – what does this mean? Was this described for the caregivers?

Results

- The authors seem to use a number of non-parametric tests, I assume this is because data are categorical? It would be helpful to include this justification.

Discussion

- This discussion was comprehensive but also a bit long. Not sure if there is a way to make this more concise but still include the information that the authors would like to convey?

- The authors include a header of “Wearable” devices, but their research questions seemed specific to smart suits and electrodes? Also, their questions regarding video and audio recordings were grouped in the results regarding issues of privacy, which seems different from whether they would adopt the use of the technology?

- I think it is also important to acknowledge the valence of caregivers’ responses. For example, while smart suits were preferred over sensing electrode stickers, the median score was 3 for sensing electrode stickers. Does this mean that participants feelings’ may be on average neutral? And perhaps with some concerns addressed they would be interested?

- I appreciate that the authors included images of the technology in conducting the survey.

- Another limitation with practicality (the authors do not need to note this in an already lengthy manuscript, but just sharing my views), is that while some technologies are meant to be easily integrated into infants’ daily lives, there are ways that caregivers adjust to the technology that may impact data collection. For example, in my own experience with LENA audio recording devices (a small recorder places inside a child’s front shirt pocket), there are times in the day when children fiddle with the recorder and remove it, or the caregiver needs to reverse the shirt so that the pocket is on the child’s back. These variations in the location of the LENA can influence what is being recorded. So, while some technologies seem easily adapted and thus fairly practical, there is still some level of maintenance on the part of the researcher to ensure the technology was appropriately adopted.

Figures

- The concept maps are interesting, but it can be hard to read some of the text when it overlaps with the nodes.

Minor

- I see that the authors acknowledge caregivers and parents on page 4 line 83, but I think it would be more inclusive of different family structures to switch to ‘caregivers’ as the main term over ‘parents’.

- Page 5 line 108 – Lena is an acronym and is generally capitalized as LENA

- Page 3 line 44: “has been made possibly” – do the authors mean “has been made possible”?

- There are a few typos throughout e.g., “user-centred”, “on-significant”, “Crochran’s Q”, “illustrative except examples”, “participating in more length investigations”, among some others.

- I believe it is more common to use “Wilcoxon” rather than “Wilcox”

Reviewer #3: This study summarizes responses to a questionnaire, sent to hundreds out households in the UK, regarding comfort with, concerns about, and orientations toward multiple types of infant developmental data collection technology that is aimed at use in the infant's home. I am happy to see that the authors have conducted this survey and think it's great that they are sharing the results. I think their work will be very useful to others thinking about collecting these types of data who are weighing the costs and benefits of more/less intrusive technologies. I have a number of small comments, but I think the manuscript is already in pretty good form:

- Please make it clear from the start that these data are limited to families in the UK. I can imagine that cultural context will influence a number of the measured outcomes, e.g., what times of day work best, what is perceived as "international", etc.

- In some cases where there is a significant difference between categories, a look at the graphical data makes apparent that, while significant, some of the differences aren't actually large in size. Since, e.g., PIs of research projects might use these data to get an idea of what a certain decision will "cost" them in terms of recruitment, it'd be useful to remind readers what significant translates to in terms of, e.g., % difference, rather than relying on the figures.

- The LENA recording device is named with an acronym ("Language ENvironment Analysis") and should be in all capital letters

- Regarding daylong audio recordings with the LENA device, a convenient overview of methodological and ethical concerns that may be useful for the current paper is Casillas & Cristia (2019; Collabra, A step-by-step guide to collecting and analyzing long-format speech environment (LFSE) recordings.) and a paper demonstrating the large shift in perspective for short vs. long at-home recordings is Bergelson et al., (2019; Dev Sci, Day by day, hour by hour: Naturalistic language input to infants). These two papers help round out some of the audio vs. video and short vs. long recording issues brought up by the authors already.

- The explanation for the preference of video over audio strikes me as somewhat unexpected given that video includes audio, could the authors please explain?

- The use of automated tools to anonymize rich data sources like audio and video are brought up, but I think need more expansion vis-a-vis the quite significant limitations they can bring, especially how accurate the tools are for any given use case (e.g., across household types, other data collection settings, cultural groups, etc.) and the extent to which validity data are needed to effectively use them.

- Regarding the previous point, if the authors have any information on how the parents understood or conceived of these automated tools, that would be useful as a standard to which these tools are expected to be held in real use cases.

- "only to be practical for shorter periods" (p. 16) >> this seems not true on the basis of the descriptive data captured in the figures. Please make a pass for these types of claims to make it clear what the distinction is between significant and meaningfully large effects, as relates to my second point.

- The "General Comments" subheader makes sense in the context of the questionnaire but always misled me. Perhaps the authors can call these "Open responses" or something along those lines?

- "Majority of stakeholders" >> It is my understanding that stakeholders includes more than the participants (e.g., the researchers and the tech developers too), so it may be useful to stick to "parents" or "guardians". Please check elsewhere for appropriate use of this term.

- "warrants investment in developing higher-grade quality versions of sensing bands/suits" >> I would like the authors to please expand a little on this. Could they propose some useful directions to go in, based on their data? (e.g., integration with custom apps via bluetooth connection, etc.)

- I found it hard to wrap my head around some of the analyses where the response categories overlap, e.g., "Day and Night", "Day", and "Night" are similar in a way that "Specific time" is not—can the authors please justify comparing these (actually) overlapping response categories as independent types?

- Finally, while I appreciate that the authors took the time to collect open responses, I found the Leximancer analyses totally unconvincing. I would advocate for removing them entirely from the paper, or reporting on individual representative examples.

Note: The manuscript cover page details state that the data are fully available without restriction, but I did not find a link to these materials in the draft.

Thanks for an interesting read!

Reviewer #4: This paper evaluates parents'/caregivers' acceptance of remote monitoring technologies to assess child development. They use three constructs to capture acceptability: 1. Practicability, 2. Privacy, 3. Feedback.

The authors' main argument is that the views parents and caregivers, the primary stakeholders, are important for the future adoption of these technologies. They note the limited research in this area (references 27 and 28). Prior research was also done in the US with a diverse population. It seems like the added value of this paper is the use of a sample from the UK and some more nuanced surveying of respondents.

The methodology is primary quantitative (i.e., survey), but they do content analysis using free text responses.

The paper would benefit from addressing methodological concerns and better structuring of their manuscript as mentioned in the comments below.

INTRODUCTION

MINOR COMMENTS

- LENA should be capitalized in line 108

- Contextualization is never defined in the introduction

METHODS

MAJOR COMMENTS

- The Methods section formatting is hard for the reader to follow. In general, it is good to have the participant recruitment/study population in the first section of the methods before mentioning the survey assessments. Further, because methods section does not align with their three constructs (practicability, privacy, feedback). Rather than having subheadings (study interaction, data sharing) under their three constructs (practicability, privacy, feedback), each is presented as headings. For example, data sharing would presumably be under privacy.

- The authors ran an extraordinary number of analyses, which though using Bonferroni corrections is concerning for false positives.

- They made their non-validated Likert responses ordinal, without displaying the number responses to participants, which means participants could ascribe differing values to the response choices. The concern, particularly given missing data, is that individual participant comparisons may not be possible.

MINOR COMMENTS

- The survey was piloted and revised, which is good

- Non-random sampling - It is worth noting that a lot of online advertising was used for recruitment, so the sample may be more tech-saavy. Also, some of the sample was from a pool of volunteers, which could also affect generalizability of results. This limitation is mentioned in part in the discussion, but primarily in regards to low SES population.

RESULTS

MAJOR COMMENTS

- Authors have a lot of figures describing demographic data where a fewer number of tables with the same data would suffice

MINOR COMMENTS

- Some of the heading formatting is off

DISCUSSION

MAJOR COMMENTS

The discussion is very long. Many results are discussed in the conclusion, which seems makes it seem unfocused. Discussing a more limited set of main findings with less numbers and percentages would be better for the reader.

6. PLOS authors have the option to publish the peer review history of their article (what does this mean?). If published, this will include your full peer review and any attached files.

Reviewer #1: No

Reviewer #2: **Yes: **Janet Y. Bang

Reviewer #3: No

Reviewer #4: No

---

## [Author Response · Author response to Decision Letter 0]

29 Sep 2020

Reviewer #1

Comment 1: My primary concern is the framing of the study appears to be centred around the needs of the particular lab for a study they are planning. However, they attempt to draw much broader conclusions that would be of interest to the larger infant research community, and the introduction sets up these themes in very broad terms. I don't believe that they have been entirely successful in drawing conclusions of relevance to the broader research community, and I don't know whether it would work to frame the study solely around the conclusions relevant to this particular lab, or even specifically labs that do smartsuit testing (it's not clear to me how many that is)?

Response: We decided to frame the questions within a concrete context to help families to give opinions that might be more closely related to how they would act in an actual study (rather than a more abstract framing). We think there is increasing interest in the developmental research community in methods for home assessment using these kinds of technologies, and this has been greatly accelerated by the current pandemic. As such, we think our results may contain insights that would be of interest to a range of labs who might be considering embarking on this path. However, we agree that our sample and framing also places some constraints on the breadth of conclusions we can draw. To be clearer, we have clarified that insights might be most relevant to researchers planning to conduct similar research in the future:

LINE 188: “…for research into psychophysiology/motor and behavioural infant development.” To specify which technologies were addressed. 

LINE 188: “Our questions were delivered in the context of real-world examples of such technologies and assessments to provide a concrete framing in which to ascertain realistic opinions from responders.” To justify why we were quite specific in our questions. 

Also see response to reviewer 3, comment 1. 

Comment 2: I selected "no" for intelligibility not being there is a problem with the clarity of the English per se, but rather because there are some structural issues that make reading challenging. Two broad concerns: 1. The sub-headers are inconsistent, making it hard to match up the research questions with the data use to test these questions. For example, it took me a fair amount of searching to figure out what the sub-section "Study Interaction" was referring to. It appears to be a sub-section of the question(s) about practicality, but is listed separately in the Analysis section. 

Response: We have aligned all headings throughout the manuscript with analysis, results and discussion structured under our initial themes as set in the introduction under study goals: Viability (subheadings: Sticker sensing electrodes versus smart suits/sensing bands, Static versus body video recording devices, Smartphones), Privacy (subheadings: Video and audio recording with/without privacy preservation, Data sharing), Data Access, and Future Participation. 

Comment 3: Relatedly, there are quite a lot of questions that are tested in a number of different ways, and the reader gets a little lost in the trees for the forest. I did not find the figures particularly helpful in this respect in visualizing the findings - the authors could take a step back and consider which aspects of the findings they most want to highlight.

Response: We have added percentages to the text in the results section to aid understand of the figures as well as to highlight the main finding. Additionally, we have amended a number of subsections to more concisely describe findings for example, in “Comparison of length of participation” results we have moved the mcnemar post-hoc text to SM. Within Privacy section, we have added to the results descriptions of percentages of those who did not change their response between the technologies to highlight the weak effects. We have also added a few sentences at the beginning of each results subsection to remind the reader of the purpose of each sub-investigation to aid understanding. 

Comment 4: The question about sharing with researchers outside of the lab does not describe any processes that might be in place to ensure participant confidentiality, which may also skew the responses toward a more conservative answer. The described purpose of the audio/video may play a key role in the participants responses. Here the survey suggests that the need for video/audio is fairly limited so participants may be more inclined to request that the data be anonymized. In my lab, participants tend to be quite willing to share their raw audio with us, as they understand that it is necessary for the research.

Response: Although we added descriptions of the technology for context, we kept the description of the purpose for all measurements relatively limited to increase generalisability, but we agree this may have influenced responses. We have added this as a point to our limitations in the manuscript. 

LINE 820: “Similarly, the descriptions in the survey for the purpose of collecting each type of data was limited, in order to improve the generalisability of our findings. However, the description of the technology may have influenced responses, which may differ with more information or because of study specifics. Information pertaining to security and participant confidentiality for questions regarding data sharing were also limited. Providing information on the protocols of data across specific collaborating labs may influence opinions on data sharing. This needs to be taken into account by future investigators.”

Reviewer #2

Comment 1: I found the structure of the paper a bit confusing where in the Introduction and Research Questions there seem to be three main areas of focus (i.e., Practicality, Privacy, and Feedback), but the Analysis plan and Results separate those themes into a few more headers, and then the discussion creates new sub headers as well. My preference would be for more alignment or a way to introduce this organization to the reader. Because there is a lot of information in this manuscript, having some consistency in structure would be helpful for the reader.

Response: See above response to Reviewer 1’s comment 2. 

Comment 2: Second, I felt like the themes raised in the Introduction were at times broader than the scope of the article. For example, the authors discuss many issues related to practicality but it seems that this is examined specifically in relation to duration of use in a day. 

Response: We agree that this was not clear. During our piloting, we noted a number of participants commenting on the length of study determining the practicality of technologies: with longer investigations being less practical. Therefore, in order to prevent ambiguity in the interpretation of the question we added to the questionnaire, specifying length of study over two questions regarding practicality. We specifically focused on reporting differing practicality rating with length of study as we believe this the most informative for the widest variety of future investigations. Additionally, we also provided open field response boxes within each practicality question for more specific concerns of practicality to be highlighted. However, we have not reported on these for conciseness of an already very comprehensive paper, as we did not identify any additional relevant themes. We have also eliminated these from the open data sharing as responders at times answered these questions non-anonymously. 

 We have added to the introduction to specify this reasoning of asking questions in this manor: LINE 255: “The survey consisted of multiple-choice questions on the following topics “ …… “The practicality of technologies over different time scales (to assess overall practicality while accounting for the possible effect length of study on practicality).”

Comment 3: The authors also raise an important issue of feedback and the potential of noticing atypical behaviours, but these concerns do not seem to be addressed in the analyses and the discussion. I would suggest specifying the scope of the paper earlier in the introduction and research questions.

Response: In response to different reviewer’s comment, we have changed the section on “feedback” to “data access”, as we believe this is a more accurate description of our questions. Because of this, as well as the angle of our questions, we have eliminating the comment on atypical behaviours in the new section “data access”. We also acknowledge that specifying the scope earlier in the paper as an important comment and we have addressed this: see our response to reviewer 1 comment 1 and reviewer 3 comment 1. 

Comment 4: The authors present this work to be important for COVID-safe measures and ‘socially-distanced’ data, which of course is extremely relevant to our times now. However, adopting in-home infant monitoring technologies are likely important for issues of including many different families who may not always have the time or access to research studies, as well as performing research at a very high scale (as the authors acknowledge). I think it would benefit the manuscript to emphasize these latter points, because the implications of this work are much broader than for the pandemic alone.

Response: We have acknowledged this comment and have added:

LINE 67: “as remote technologies can be implemented by caregivers, they also provide an opportunity for socially distanced data collection at a more scalable level and may facilitate the participation of participant groups who would find it difficult to come to a lab (e.g. those with disabilities, working caregivers or remotely located families).”

Comment 5: the authors seem to focus on specific comparisons between technologies in their analyses, which are not immediately obvious to the reader. It would be helpful to include this specificity in their research questions.

Response: The comparison of technologies falls into broad categories of low or high active care (stickers are high active care as they require more time and technique to apply, as described in the introduction paragraph titled “viability”. We wanted to compare across the levels of active care within particular data types (e.g. physiology or video capture of behaviour). We have reworded some of the viability subsection of the introduction for clarity. See below for examples:

LINE 198: “For video data, are static low active care recording devices preferable to all-encompassing high active care on-body video recording devices?”

LINE 100: “In terms of viability, remote tools and assessments can be categorised by the required amount of active care.”

LINE 103: “The latter can be easily and quickly put on/taken off the infant and are not too far removed from daily dressing routines and are therefore low active care devices.”

LINE 114: “Despite this, caregivers’ opinions on static low active cares versus on-body high active care video devices is unclear”

Additionally, for clarification of this we have added to our viability study research questions: 

LINE 194: “For the collection of infant psychophysiological/motor data, are lower active care infant wearable technologies (e.g. smart suits and wrist/ankle bands) preferable to more traditional higher active care technologies (e.g. sensing electrode stickers)?”

LINE 198: “Are static low active care recording devices preferable to all-encompassing high active care on-body video recording devices?”

Comment 6: Page 4 line 69: “Although user-friendly, commercial devices allow limited access to raw, high resolution data and are therefore poorly equipped for more complex research questions” – I don’t understand why high-resolution data is poorly equipped for more complex research questions? It seems that high-resolution data would in fact be great for more complex research questions?

Response: We apologise this wasn’t clear – we meant that access to that kind of data is often not possible or highly limited (rather than that that type of data wouldn’t be useful) – we have re-worded this:

LINE 73: “User-friendly, commercial devices do not allow access to raw, high resolution data and are therefore poorly equipped for more complex research questions”.

Comment 7: The authors discuss practicality here in many ways, but the analyses seem to focus on duration of use in a day? I suggest to specify that focus here to help orient the reader. The terminology of practicality and practicability are also used interchangeably, I would suggest using one for consistency unless they convey different meanings.

Response: See response to comment 1. Also, Practicability means whether it is “viable” or “possible” for caregivers. Practicality is whether it is easy or efficient. We acknowledge the similarity of these two words, so we have changed the use of Practicability to Viability so that the reader can easily distinguish the points we are making,

Comment 8: I’m not sure I understand the paragraph on the ‘contextualization’ of data (starting line 114). Aren’t all in-home technologies providing the contextualization of a more ecological setting?

Response: We agree that home-based technologies are designed to enable capture in a natural context. We meant to convey that researchers may want additional information about the nature of that context. Homes will naturally vary more than a single lab setting, so additional measures of that context may be needed to aid interpretation. For example, for physiological data we might want to ask how changes in an infant’s heart rate are coupled with variation in noise levels in their environment. Contextualisation in this sense is being used to refer to the video recording or parent reports on a smart phone that can be used to understand what is happening in the environment. 

We have amended the paragraph for clarity:

LINE 120: “The contextualisation of data is another key requirement of remote designs, particularly for relatively lower resolution data (e.g. physiological measures that do not provide identifiable data). In a lab setting, environments are often controlled but, in a home setting the context in which the data is being collected can vary substantially (e.g. dinner time, playtime, bedtime etc). Recording this contextual information is sometimes required in order to understand the data (e.g. did an infant’s heartrate increase because they were crying or laughing?). The increasing societal ubiquity of smartphones (33) make them an ideal tool for the collection of both contextualisation data and primary data (33,34)”

Comment 9: The authors mention the use of resolution in a couple of places but it doesn’t seem consistent (e.g., line 69 high resolution data vs. line 115 lower resolution data like physiological measures). Are physiological measures considered lower resolution because they are not identifiable like audio or video recordings?

Response: Yes this is what we meant. But this sentence has now be redacted and our point has been clarified within the new contextualisation paragraph starting from LINE 120. 

Comment 10: Rather than “Feedback”, perhaps using a different term specific to families’ access to the data is better? It doesn’t seem like the authors examine findings related to the issues of caregiver concerns of atypical behaviour.

Response: We have changed “feedback” to “data access”.

Comment 11: How were families compensated?

Response: We have added:

LINE 242: “Participants received no monetary compensation for their time.” To the end of the “recruitment procedure” section of the methods.

Comment 12: Do the authors have information on how long it took caregivers to respond to the surveys (e.g., mean and range)?

Response: We only have the average as this is the only statistic from survey monkey – we have added:

LINE 246: “which took an average of 12 minutes 18 seconds to complete.”

Comment 13: I had difficulty understanding this sentence on page 12 lines 260 – 263: “Note that all McNemar tests in this paper conducted for each categorical option on 2*2 contingency tables constructed with number of responders selecting that categorical option versus number of responders not selecting that category across the items being assessed e.g. technology.”

Response: We added this sentence to clarify the sentence before, but after revisions we are aware this may have confused the reader. We have opted to remove this sentence and added to the previous sentence so that it now reads: 

LINE 295: “We used Bhapkar tests with 2*2 contingency Bonferroni corrected McNemar post-hoc test to compare selecting versus not selecting the response category between technologies for each response separately.”

Comment 14: Is there a table that could help readers understand the different technologies targeted in the survey?

Response: We have added this: 

LINE 249: “Sections 2-7 asked about the responder’s attitudes to the following technologies: smart suits, sensing electrode stickers, wrist/ankle bands, video recording, audio recording, and smartphones (see S4 Table A for summary of technologies)”. 

Comment 15: “Now and Then” – what does this mean? Was this described for the caregivers?

Response: Now and then is a common phrase meaning occasionally in British English, this was not explained to participants but we assumed our British sample would be accustomed to the phrase and its meaning.https://www.collinsdictionary.com/dictionary/english/now-and-then

Comment 16: The authors seem to use a number of non-parametric tests, I assume this is because data are categorical? It would be helpful to include this justification.

Response: Yes, as the data was categorical and ordinal. We have added:

LINE 276: “using non-parametric tests due to the ordinal and categorical nature of the data.” to the analysis section for clarity.

Comment 17: This discussion was comprehensive but also a bit long. Not sure if there is a way to make this more concise but still include the information that the authors would like to convey?

We have streamlined the discussion by eliminating numbers and percentages (as these are merely reiterations from the results). We have also revised throughout to improve concision, shortening lengthy sentences and removing excerpts. To streamline the points make in the discussion, we have also removed the point within the Privacy subheading on the difference between video and audio on length of study as after review this point was overly elaborate connection between the results and the open response and potentially inflate the relatively small effect size of the results. 

Comment 18: The authors include a header of “Wearable” devices, but their research questions seemed specific to smart suits and electrodes? Also, their questions regarding video and audio recordings were grouped in the results regarding issues of privacy, which seems different from whether they would adopt the use of the technology?

Response: See above response to Reviewer 1’s comment 2. Additionally, we have grouped video/audio technologies under privacy as the primary concern families are likely to have with these modalities concerns privacy (rather than practicality – as is demonstrated by themes of privacy in the open responses to audio and video). Furthermore, we compared these technologies with and without privacy preserving techniques, providing further justification to group in the privacy section to prevent reptation of analysis. 

Comment 19: I think it is also important to acknowledge the valence of caregivers’ responses. For example, while smart suits were preferred over sensing electrode stickers, the median score was 3 for sensing electrode stickers. Does this mean that participants feelings’ may be on average neutral? And perhaps with some concerns addressed they would be interested?

Response: We have considered this comment and added to the discussion:

LINE 512: “It must be noted that although bands/body suits were rated more favourably in the qualitative rating, stickers were viewed neutrally by caregivers (selecting a median score of 3/neutral on the rating scale). Potentially, by addressing the safety concerns highlighted in the open responses, stickers may be seen more favourably.”

Comment 20: The concept maps are interesting, but it can be hard to read some of the text when it overlaps with the nodes.

Response: Unfortunately, this is the way the Leximancer software creates the maps. We tried to make it as legible as possible by adjusting the character size. We believe that although some words overlap slightly, in our opinion all are readable as different colours have been used. 

Comment 21: I see that the authors acknowledge caregivers and parents on page 4 line 83, but I think it would be more inclusive of different family structures to switch to ‘caregivers’ as the main term over ‘parents’

Response: All “parents” are changed to “caregiver” other than those that were included in the illustrative quotes from responders. 

Comment 22: Page 5 line 108 – Lena is an acronym and is generally capitalized as LENA

Response: lena changed to LENA

Comment 23: Page 3 line 44: “has been made possibly” – do the authors mean “has been made possible”?

Response: Corrected typo to “possible”

Comment 34: There are a few typos throughout e.g., “user-centred”, “on-significant”, “Crochran’s Q”, “illustrative except examples”, “participating in more length investigations”, among some others.

Response: We have amended these that have been noted and amended others throughout (as seen in the tracked changes document) 

Comment 35: I believe it is more common to use “Wilcoxon” rather than “Wilcox”

Response: This typo has been corrected 

Reviewer #3: 

Comment 1: Please make it clear from the start that these data are limited to families in the UK. I can imagine that cultural context will influence a number of the measured outcomes, e.g., what times of day work best, what is perceived as "international", etc.

Response: We have revised the text to emphasise that the insights may be most relevant to the UK, and their cultural generalisability should be explored: 

Line 92-96. We state our rational that there hasn’t been any canvassing of the opinions of a UK cohort and this is our goal. 

Line 180: we state our goal is to canvass opions of a UK caregivers. I have clarified the potential of the research to provide guidance to (line 182) “similar UK based lab/research groups”

IINE XXX : “UK caregiver” to clarify our aim. 

Line 562 – 564: we have added “a Uk based cohort”, “with similar aims”

Comment 2: In some cases where there is a significant difference between categories, a look at the graphical data makes apparent that, while significant, some of the differences aren't actually large in size. Since, e.g., PIs of research projects might use these data to get an idea of what a certain decision will "cost" them in terms of recruitment, it'd be useful to remind readers what significant translates to in terms of, e.g., % difference, rather than relying on the figures.

Response: We have noted this and added percentages to the results section to highlight our main findings. We have also reiterated small effect sizes by acknowledging the percentages of response that do not change in the results and highlighting this in the discussion. 

 Addition of percentages can be noted throughout results.

 Addition of clarity to the small effect sizes can be notes by the following additions.

 Results:

LINE 460: “Respectively, 51.66%, 55.37% and 55.75% of caregivers did not consider privacy preserving technique to change their likelihood of participation with on-body camera, cot camera and audio recording” 

LINE 476: Comparison of practicality. “Averaged across both lengths of time, 58.31% of participants indicated video and audio as equally practical (Fig 2B).”

LINE 481: “Though this was to a weak effect (Kendall W = 0.227), with 47.21% and 42.40% of the sample indicating “Short Period” of recording more practical than “Extended Period” for video and audio respectively.”

LINE 490: “Averaged across recording technology, 66.67% of responders did not change their response when given the privacy preserving option.”

Discussion:

LINE 677: “did increase favourability of technology but to a moderate degrees, with the approximately half of participants responding the same with and without the option of privacy preserving processing within technology”

LINE 688: “The lack of preference to either video or audio, as well as the minimal favourability to privacy preserving processing was reiterated in responses to preferred study duration. The majority of participants did not change their response between technologies nor between privacy preserving option.”

Comment 3: The LENA recording device is named with an acronym ("Language ENvironment Analysis") and should be in all capital letters

Response: lena changed to LENA

Comment 4: Regarding daylong audio recordings with the LENA device, a convenient overview of methodological and ethical concerns that may be useful for the current paper is Casillas & Cristia (2019; Collabra, A step-by-step guide to collecting and analyzing long-format speech environment (LFSE) recordings.) and a paper demonstrating the large shift in perspective for short vs. long at-home recordings is Bergelson et al., (2019; Dev Sci, Day by day, hour by hour: Naturalistic language input to infants). These two papers help round out some of the audio vs. video and short vs. long recording issues brought up by the authors already.

Response: We agree that here two papers are very relevant to our manuscript and have added both of these reference to our paper: 

 LINE 65: “Similarly, longer measurement periods may also shed new light on behaviours that have traditionally been measured over relatively short epochs of data collection (12,13)”

Comment 5: The explanation for the preference of video over audio strikes me as somewhat unexpected given that video includes audio, could the authors please explain?

Response: Video does not necessarily always include audio – in our survey we were questioning moving image recording only (video without audio) and audio separately. As we have different responses across the two we assume participants interpreted this in this way. Nonetheless we have added to our discussion the ambiguity of this questioning. 

Comment 6: The use of automated tools to anonymize rich data sources like audio and video are brought up, but I think need more expansion vis-a-vis the quite significant limitations they can bring, especially how accurate the tools are for any given use case (e.g., across household types, other data collection settings, cultural groups, etc.) and the extent to which validity data are needed to effectively use them.

Response: We have addressed this comment and added it to the discussion . 

LINE 701: “Researchers must therefore thoroughly consider beforehand what variables are needed for analysis, and the validity of extracted data using the chosen method given the data collection setting (e.g. will the infant’s vocal pitch be validly extracted if collected in a noisy household).”.

We have also alluded to the privacy-preserving technique having pitfalls in the introduction LINE 148: “Considering how participants view the efficacy of privacy-preserving measures in the context of different research designs will enable researchers to make informed decisions on the trade-off between data quality and participant uptake/attrition.” 

Comment 7: Regarding the previous point, if the authors have any information on how the parents understood or conceived of these automated tools, that would be useful as a standard to which these tools are expected to be held in real use cases.

Response: We agree that this would be really interesting. Unfortunately, we do not have any information on this, but we have noted the impact of differing perceptions/understanding of the technology on the opinion in the discussion:

LINE 822: “the description of the technology may have influenced responses, which may differ with more information or because study specifics.”

LINE 826: “Ultimately, the level of participant understanding of the technology/assessment specific is likely to influence opinion, thus, needs to be taken into account by future investigators.”

Comment 8: - "only to be practical for shorter periods" (p. 16) >> this seems not true on the basis of the descriptive data captured in the figures. Please make a pass for these types of claims to make it clear what the distinction is between significant and meaningfully large effects, as relates to my second point.

Response: We are aware that the wording of this comment may have been misleading. We have now changed this to “would be more preferred for shorter periods”. We have also gone through the whole manuscript and made sure that the interpretations are consistent with the effect sizes on the graph. And have added percentages to the results section to reiterate this. See response to comment 2. 

Comment 9: The "General Comments" subheader makes sense in the context of the questionnaire but always misled me. Perhaps the authors can call these "Open responses" or something along those lines?

Response: We have changed all “general comments” to “open-ended responses”

Comment 10: "Majority of stakeholders" >> It is my understanding that stakeholders includes more than the participants (e.g., the researchers and the tech developers too), so it may be useful to stick to "parents" or "guardians". Please check elsewhere for appropriate use of this term.

Response: Changed all “stakeholders” to “caregivers”

Comment 11: - "warrants investment in developing higher-grade quality versions of sensing bands/suits" >> I would like the authors to please expand a little on this. Could they propose some useful directions to go in, based on their data? (e.g., integration with custom apps via bluetooth connection, etc.)

Response In terms of development we were considering more generally, and have added an example to the end of the sentence the reviewer mentioned:

LINE 586: “e.g. sensors wirelessly connected to apps that would pre-process data remotely, and upload anonymised derived data.”. 

Comment 12: I found it hard to wrap my head around some of the analyses where the response categories overlap, e.g., "Day and Night", "Day", and "Night" are similar in a way that "Specific time" is not—can the authors please justify comparing these (actually) overlapping response categories as independent types?

Response: We included response options in this manner, as “Day and Night” refers to the total 24hour period of day and night. Whereas “Day” and “Night” refer to only the day period or only the night period, but not both. We believe the way our participants interpreted it as there was a difference as it was a forced choice (i.e. the could not choose both “day and night” and “night”). 

Comment 13: Finally, while I appreciate that the authors took the time to collect open responses, I found the Leximancer analyses totally unconvincing. I would advocate for removing them entirely from the paper, or reporting on individual representative examples.

Response: We acknowledge the shortcomings of the Leximancer approach in the limitation section of the discussion. 

LINE 811: “We attempted to conduct a rudimentary qualitative analysis using topographic maps that extrapolated themes/concepts based on the number of mentions. However, this was not an exhaustive systematic analysis of the qualitative data, qualifying the depth of conclusions we can draw.”. 

LINE 589: “(though it is important to note that these represent a simple summation of the most common words used, and future work should employ more extensive qualitative methods).”

We have chosen not to report on individual representative examples because we are concerned that would run the risk of bias or misrepresenting our data. We want to retain this data in the paper as we believe it valuable to understanding the motivation behind quantitative findings and highlightly themes/concepts that we had not considered. A notion we have added to the introduction. 

LINE 278: “This analysis was not intended to be exhaustive, but to highlight themes beyond those considered when designing the survey.”

Comment 14: : The manuscript cover page details state that the data are fully available without restriction, but I did not find a link to these materials in the draft.

Response: We added the link to the PLOS one submission. Unsure why this reviewer had an issue with finding it? 

Reviewer #4

Comment 1: LENA should be capitalized in line 108

Response: lena changed to LENA

Comment 2: Contextualization is never defined in the introduction 

Response: Added the following for clarity:

 LINE 120: The contextualisation of data is another key requirement of remote designs (e.g. physiological measures that do not provide identifiable data). In a lab setting environments are controlled, but in a home setting the context in which the data is being collected can vary dramatically both within and between infants (e.g. dinner time, playtime, bedtime etc). Recording this contextual information is sometimes required in order to understand the data (e.g. did an infant’s heartrate increase because they were crying or laughing?). The increasing societal ubiquity of smartphones (33) make them an ideal tool for the collection of both contextualisation data and primary data (33,34).

Comment 3: The Methods section formatting is hard for the reader to follow. In general, it is good to have the participant recruitment/study population in the first section of the methods before mentioning the survey assessments. Further, because methods section does not align with their three constructs (practicability, privacy, feedback). Rather than having subheadings (study interaction, data sharing) under their three constructs (practicability, privacy, feedback), each is presented as headings. For example, data sharing would presumably be under privacy.

Response: We have reformatted the manuscript methods section to start with participants, where the sample characteristics have been described. This is followed by recruitment and then the survey. We have aligned all headings throughout the manuscript with analysis, results and discussion structured under our initial themes as set in the introduction under study goals.

Comment 4: The authors ran an extraordinary number of analyses, which though using Bonferroni corrections is concerning for false positives.

Response: We have noted this in the limitations section. 

 LINE 829: “In order to comprehensively cavass caregivers’ opinions on remote monitoring technologies and study designs, we asked a large number of question. Consequently, we conducted a large number of comparisons for which we attempted to correct for multiple comparisons within each analysis using Bonferroni correction. Nonetheless there is the potential for false positives. Therefore, we present all analysis alongside appropriate effects size and graphs, as well as made the data available. This is to enable future investigators to evaluate the information that be most relevant to their particular study.”

Comment 5: They made their non-validated Likert responses ordinal, without displaying the number responses to participants, which means participants could ascribe differing values to the response choices. The concern, particularly given missing data, is that individual participant comparisons may not be possible.

Response: The response options were not presented to the participants in a numerical likert scale as we considered separate qualitative options e.g. “Extremely likely”, to reduce likelihood of participants ascribing differing meaning to the answer. Retrospectively, we are aware that different participants may ascribe different differences between the responses, as they were not numerically scaled. However, most comparisons were conducted within subject, therefore we believe should reduce the possibility of this influencing the results as within individual’s are likely to use the same construct. 

Comment 6: Non-random sampling - It is worth noting that a lot of online advertising was used for recruitment, so the sample may be more tech-saavy. Also, some of the sample was from a pool of volunteers, which could also affect generalizability of results. This limitation is mentioned in part in the discussion, but primarily in regards to low SES population.

Response: We have added to our limitations to make this point more explicit. 

LINE 794: “, thus people who responded to the survey were probably those already likely to engage in and would have an already high level of acceptance of our research, limiting the scope of generalisation to a degree. The recruitment stream used may have particularly limited access to low SES families, as noticed in the demographic of our sample.”

Comment 7: Authors have a lot of figures describing demographic data where a fewer number of tables with the same data would suffice

Response: This is now in a table

Comment 8: Some of the heading formatting is off

Response: We have corrected this

Comment 9: The discussion is very long. Many results are discussed in the conclusion, which makes it seem unfocused. Discussing a more limited set of main findings with less numbers and percentages would be better for the reader.

Response: see response to reviewers 2 comment 7. 

Editor response:

Comment 1: Please ensure that your manuscript meets PLOS ONE's style requirements, including those for file naming. The PLOS ONE style templates can be found at https://journals.plos.org/plosone/s/file?id=wjVg/PLOSOne_formatting_sample_main_body.pdf and https://journals.plos.org/plosone/s/file?id=ba62/PLOSOne_formatting_sample_title_authors_affiliations.pdf

Response: we have considered the mentioned links and believe we have followed the guidance provided. We have amended 

Comment 2: Please amend either the title on the online submission form (via Edit Submission) or the title in the manuscript so that they are identical.

Response: we have aligned both title pages.

Comment 3: Please include additional information regarding the survey or questionnaire used in the study and ensure that you have provided sufficient details that others could replicate the analyses. For instance, if you developed a questionnaire as part of this study and it is not under a copyright license more restrictive than CC-BY, please include a copy, in both the original language and English, as Supporting Information

Response: we have added to the manuscript that the questionnaire was developed for the purpose of the study. 

 LINE 245: “We developed the survey for the purpose of this study (See S1 for the full non-copyright survey).”

Comment 4: Please note that PLOS ONE uses a single-blind peer review procedure. We would therefore be grateful if you could include in the information that has been redacted for peer review in the manuscript.

Response: We have included all identifiable information. 

Kind regards, 

Laurel Fish

---

## [Decision Letter · Decision Letter 1]

11 Nov 2020

PONE-D-20-20163R1

A survey on the attitudes of parents with young children on in-home monitoring technologies and study designs for infant research.

PLOS ONE

Dear Drs. Fisher,

Thank you for submitting your manuscript to PLOS ONE. After careful consideration, we feel that it has merit but does not fully meet PLOS ONE’s publication criteria as it currently stands. Therefore, we invite you to submit a revised version of the manuscript that addresses the points raised during the review process. Overall, as you will see reviewers were quite satisfied with your revised manuscript, but had some additional remarks to further improve upon it. Please try to incorporate these as much as possible in the revised version of your manuscript, 

We look forward to receiving your revised manuscript.

Kind regards,

Barbara Schouten

Academic Editor

PLOS ONE

Reviewers' comments:

Reviewer's Responses to Questions

**Comments to the Author**

1. If the authors have adequately addressed your comments raised in a previous round of review and you feel that this manuscript is now acceptable for publication, you may indicate that here to bypass the “Comments to the Author” section, enter your conflict of interest statement in the “Confidential to Editor” section, and submit your "Accept" recommendation.

Reviewer #1: (No Response)

Reviewer #2: (No Response)

Reviewer #3: (No Response)

Reviewer #4: (No Response)

2. Is the manuscript technically sound, and do the data support the conclusions?

Reviewer #1: Partly

Reviewer #2: Yes

Reviewer #3: Yes

Reviewer #4: Partly

3. Has the statistical analysis been performed appropriately and rigorously? 

Reviewer #1: Yes

Reviewer #2: I Don't Know

Reviewer #3: Yes

Reviewer #4: Yes

4. Have the authors made all data underlying the findings in their manuscript fully available?

Reviewer #1: Yes

Reviewer #2: Yes

Reviewer #3: Yes

Reviewer #4: Yes

5. Is the manuscript presented in an intelligible fashion and written in standard English?

Reviewer #1: Yes

Reviewer #2: Yes

Reviewer #3: Yes

Reviewer #4: Yes

6. Review Comments to the Author

Reviewer #1: The structure of the paper is much more readable, and the authors have been very responsive to prior critiques. I still feel that some of the conclusions are a little broad given the limitations of the survey. I appreciate the need to give specific examples, but I still feel that this may skew the responding in ways that are not known. They have added some wording to acknowledge this, however, and the findings more generally would be of interest to a broader audience even if care must be taken in generalizing these findings to other labs.

Reviewer #2: I appreciate the author’s detailed responses to the comments. I think this revision is an improvement, with clearer goals and in a more concise and organized format. I believe that this will be an interesting contribution to the literature. Many of my comments have been addressed appropriately. I have a few more comments below:

Intro

- Unclear how smartphones are helping the contextualization? The explanation here doesn’t address contextualization in the ways addressed earlier in the paragraph.

Methods

- Lines 254 – 259 seem like a list? I think this would be better set up as an enumerated list or with a colon or semi-colons to group these together.

Results

- It seems like there are results presented here that don’t follow from the research questions? For example, the authors pose research questions as related to 1) preferences of lower active vs. higher active technologies, 2) static vs. all-encompassing technologies, and 3) smartphones, but the research questions are framed as preferences and without the different analytic headers as seen in the results re likelihood practicality, and duration with different technologies (although I see the Results and Discussion headers follow consistently).

- Additionally, in the research question the authors mention write/ankle bands, but in the Results we are routed to the supplemental material – making that consistent from the beginning would be helpful (I may be misunderstanding this?). They also use the term sensing bands here in the header – which seems to encompass wrist/ankle bands? The organization in the Results in relation to the Research Questions was still a bit confusing for me.

- I have no comment on the use of “Bhapkar tests with a 2*2 contingency Bonferroni corrected McNemar post-hoc test” “. I am just unfamiliar with this test.

Reviewer #3: I am overall satisfied with the revisions and I thank the authors for their changes. I have only a few minor remaining comments:

- Without revealing anything about this study I contacted a British English speaking colleague as well as a non-native English-speaking colleague about what "video" means to them in this context. While the BrEng colleague's judgment fell well in line with the authors' the non-native English speaking colleague's judgments fell in line with my own: that "video" without further specification is ambiguous as to an image-only stream or an audio-visual stream. Please add just a very short clarification to avoid dialectical issues in interpretation.

- There is still at least one case of "stakeholder" that needs to be replaced: "it is critical to gather stakeholder’s views" >> should be "caregivers'" or perhaps "participating families'"; please check again for other cases.

- The context for sending the weblink comes after it's first mentioned (i.e., after information about participant responses), which may be confusing. Could the recruitment information come first?

- Please mention "UK" again at the top of the summary as it's appropriate to specify scope in a discussion section

- There are a number of minor typos/grammatical errors throughout the manuscript (e.g., some possessive apostrophes, extra capitalization, extra/missing spaces).

Thanks!

Reviewer #4: INTRODUCTION

Minor comment, Line 122: This sentence is slightly confusing as written. Adding a comma after "lab setting" would make it more readable. "In a lab setting environments are controlled, but in a home setting the context in which the data is being collected can vary dramatically both between infants (e.g. single child, multi-generational household, etc) and within infant across a day (e.g. dinner time, playtime, bedtime etc)."

Minor comment, Line 144: Please revise this sentence for clarity: " A previous parent opinion survey established privacy-preserving techniques (e.g. the implementation of computer algorithms to automatically extract behaviour markers independent of identity), to only minorly improve willingness to participate in the collection of identifiable measures."

METHODS:

Minor comment, Line 222: Some people may oppose the demographic results being in the methods section as opposed to the results section. Please make sure this aligns with the journal guidelines

Minor comment: Is the abbreviation B.A.M.E necessary in Table 1? Can this be spelled out?

Minor comment, Line 278: Please add a reference for researchers who may want replicate the text mining analysis. In the supplement there are reference citations, but I do not see actual references.

RESULTS:

Minor comment: Some sentences start with numerical percentages. Please make sure this aligns with journal guidelines.

DISCUSSION:

Major comment: Given the limitations of this text mining analysis, it would be nice to put the results about caregivers' concerns for safety in the context of the existing literature. Is there literature finding similar or different results regarding safety?

Minor comment, Line 830: Canvass is misspelled: "In order to comprehensively cavass caregivers’ opinions on remote monitoring technologies and study designs, we asked a large number of question."

7. PLOS authors have the option to publish the peer review history of their article (what does this mean?). If published, this will include your full peer review and any attached files.

Reviewer #1: No

Reviewer #2: No

Reviewer #3: No

Reviewer #4: No

---

## [Author Response · Author response to Decision Letter 1]

21 Dec 2020

Reviewer #1: 

The structure of the paper is much more readable, and the authors have been very responsive to prior critiques. I still feel that some of the conclusions are a little broad given the limitations of the survey. I appreciate the need to give specific examples, but I still feel that this may skew the responding in ways that are not known. They have added some wording to acknowledge this, however, and the findings more generally would be of interest to a broader audience even if care must be taken in generalizing these findings to other labs.

Response: We have added to our conclusions “Although generalisability should be considered,” to re-highlight this limitation (line 1146)

Reviewer #2: 

I appreciate the author’s detailed responses to the comments. I think this revision is an improvement, with clearer goals and in a more concise and organized format. I believe that this will be an interesting contribution to the literature. Many of my comments have been addressed appropriately. I have a few more comments below:

Comment 1: Intro: Unclear how smartphones are helping the contextualization? The explanation here doesn’t address contextualization in the ways addressed earlier in the paragraph.

Response: We have added: “Contextualising information could be collected using smartphones via self-reporting (37). For example, a caregiver could input infant’s current activity in response to a prompt.” (lines 136 – 138)

Comment 2: Methods: Lines 254 – 259 seem like a list? I think this would be better set up as an enumerated list or with a colon or semi-colons to group these together.

Response: we have added colon and semi-colon to group this together. 

Comment 3: Results: It seems like there are results presented here that don’t follow from the research questions? For example, the authors pose research questions as related to 1) preferences of lower active vs. higher active technologies, 2) static vs. all-encompassing technologies, and 3) smartphones, but the research questions are framed as preferences and without the different analytic headers as seen in the results re likelihood practicality, and duration with different technologies (although I see the Results and Discussion headers follow consistently).

Response: We have added subheadings such as “Sticker sensing electrodes versus smart suits” into both the study goals section and the methods that are consistent with the results and discussion to make this clearer. We’ve also reworded some of the method section under these subheadings to make it clearer which questions (e.g. likelihood, practicality and length of participation) were asked in each section. 

Comment 3: Additionally, in the research question the authors mention write/ankle bands, but in the Results we are routed to the supplemental material – making that consistent from the beginning would be helpful (I may be misunderstanding this?). They also use the term sensing bands here in the header – which seems to encompass wrist/ankle bands? The organization in the Results in relation to the Research Questions was still a bit confusing for me.

Response: We wanted to point the reader to the SM for this analysis as the paper is already very long as previous comments on this paper have pointed out. Also the question/findings from this analysis are similar to the stickers v smart suits so they are more complementary, so they were put in the SM. To make this clearer we’ve removed “sensing bands” from the subheadings and research question, and have “Similar analysis comparing smart suits and sensing electrode stickers to wrist/ankle bands was also conducted (Methods and Results are reported in S6).”, in the methods section’s analysis section under Sticker sensing electrodes versus smart suits, so that it is more of a sub/complementary analysis.

Reviewer #3

 I am overall satisfied with the revisions and I thank the authors for their changes. I have only a few minor remaining comments:

Comment 1: Without revealing anything about this study I contacted a British English speaking colleague as well as a non-native English-speaking colleague about what "video" means to them in this context. While the BrEng colleague's judgment fell well in line with the authors' the non-native English speaking colleague's judgments fell in line with my own: that "video" without further specification is ambiguous as to an image-only stream or an audio-visual stream. Please add just a very short clarification to avoid dialectical issues in interpretation.

Response: We have added “image-only video” where appropriate throughout the paper:

- Aims: 

- Privacy section of methods: 

- Results section: “Static versus body image-only video recording devices”.

- Results Privacy section overview paragraph: “In-home remote infant monitoring studies, particularly those with highly identifiable image-only video and audio recording, raise privacy issues.”

- Discussion: “Static versus body image-only video recording devices” section title and we added to the first line: “Caregivers indicated similar levels of usage likelihood for static cot cameras versus on-body camera for image-only video recording”

- Discussion: “Image-only video versus audio recording with/without privacy preservation” section title. “And Similar to smartphones, audio or image-only video recording not only contribute contextualising data, but additional informative data”…. “The lack of preference to either image-only video or audio”

- 

Comment 2: There is still at least one case of "stakeholder" that needs to be replaced: "it is critical to gather stakeholder’s views" >> should be "caregivers'" or perhaps "participating families'"; please check again for other cases.

Response: Corrected to caregivers

Comment 3: The context for sending the weblink comes after it's first mentioned (i.e., after information about participant responses), which may be confusing. Could the recruitment information come first?

Response: We have moved the Recruitment procedure to come before Participants section.

Comment 4: Please mention "UK" again at the top of the summary as it's appropriate to specify scope in a discussion section

Response: Added “in the UK” to Fig 9. Description. And “ in-home monitoring with infant participants in the UK” to the figure title. 

Comment 5: There are a number of minor typos/grammatical errors throughout the manuscript (e.g., some possessive apostrophes, extra capitalization, extra/missing spaces).

Response: With careful review, we have now corrected all those we are aware of. 

Reviewer #4

Comment 1: INTRODUCTION: Minor comment, Line 122: This sentence is slightly confusing as written. Adding a comma after "lab setting" would make it more readable. "In a lab setting environments are controlled, but in a home setting the context in which the data is being collected can vary dramatically both between infants (e.g. single child, multi-generational household, etc) and within infant across a day (e.g. dinner time, playtime, bedtime etc)."

Response: The comma has been added. 

Comment 2: Minor comment, Line 144: Please revise this sentence for clarity: " A previous parent opinion survey established privacy-preserving techniques (e.g. the implementation of computer algorithms to automatically extract behaviour markers independent of identity), to only minorly improve willingness to participate in the collection of identifiable measures."

Response: We have reviewed this sentence and hope it is now clear. “A recent report on a USA-based sample of parents indicated privacy-preserving techniques to minorly improve willingness to participate in the collection of identifiable data on their infant (29). Such privacy-preserving techniques included the implementation of computer algorithms to automatically extract measures of behaviour and remove identifiable information (29).”

Comment 3: METHODS: Minor comment, Line 222: Some people may oppose the demographic results being in the methods section as opposed to the results section. Please make sure this aligns with the journal guidelines

Response: We have divided the “Participants” section. In the methods we explain how many people participated and refer to the attrition SM. The demographics part of that paragraph has now been moved to the beginning of the results section under “Sample Demographics” 

Comment 4: Minor comment: Is the abbreviation B.A.M.E necessary in Table 1? Can this be spelled out?

Response: We have changed this to “Black, Asian and Minority Ethnic Group”. 

Comment 5: Minor comment, Line 278: Please add a reference for researchers who may want replicate the text mining analysis. In the supplement there are reference citations, but I do not see actual references.

Response: Reference has been added (Smith A. Leximancer Pty Ltd [Internet]. Brisbane, Australia; 2009. Available from: https://info.leximancer.com)

Comment 6: RESULTS: Minor comment: Some sentences start with numerical percentages. Please make sure this aligns with journal guidelines.

Response: We have now corrected this throughout the paper. 

Comment 7: DISCUSSION: Major comment: Given the limitations of this text mining analysis, it would be nice to put the results about caregivers' concerns for safety in the context of the existing literature. Is there literature finding similar or different results regarding safety?

Response: This has been noted in previous literature, we have added (line 868) “This finding was consistent with a previous qualitative report on potential barriers for participation with ambulatory infant sensing devices; caregivers expressed concerns about the comfort of the physical placement of the sensor on their child (29). Though such concerns are likely to be addressed by manufacturers of the technologies as well as local ethics boards prior to implementation, future researchers should explicitly state the safety of these measures during advertising/consent to improve participant uptake.”

Comment 8: Minor comment, Line 830: Canvass is misspelled: "In order to comprehensively cavass caregivers’ opinions on remote monitoring technologies and study designs, we asked a large number of question."

Response: Corrected.

---

## [Editor Report · Decision Letter 2]

8 Jan 2021

A survey on the attitudes of parents with young children on in-home monitoring technologies and study designs for infant research.

PONE-D-20-20163R2

Dear Dr. Fish,

We’re pleased to inform you that your manuscript has been judged scientifically suitable for publication and will be formally accepted for publication once it meets all outstanding technical requirements.

Kind regards,

Barbara Schouten

Academic Editor

PLOS ONE

---

## [Editor Report · Acceptance letter]

18 Jan 2021

PONE-D-20-20163R2 

A survey on the attitudes of parents with young children on in-home monitoring technologies and study designs for infant research. 

Dear Dr. Fish:

I'm pleased to inform you that your manuscript has been deemed suitable for publication in PLOS ONE. Congratulations! Your manuscript is now with our production department. 

Kind regards, 

on behalf of

Dr. Barbara Schouten 

Academic Editor

PLOS ONE